# Spike-based Digital Brain: a novel fundamental model for brain activity analysis

**Shaolong Wei**[1], **Qiyu Sun**[2], **Mingliang Wang**[2*], **Liang Sun**[3], **Weiping Ding**[1], **Jiashuang Huang**[1*]

[1]School of Artificial Intelligence and Computer Science, Nantong University
[2]School of Computer Science, Nanjing University of Information Science and Technology
[3]College of Artificial Intelligence, Nanjing University of Aeronautics and Astronautics
`weishaolong37@gmail.com, sunqiyu0210@163.com,`
`{wml489,sunl}@nuaa.edu.cn, {dwp9988,hjshdym}@163.com`

## Abstract

Modeling the temporal dynamics of the human brain remains a core challenge in computational neuroscience and artificial intelligence. Traditional methods often ignore the biological spike characteristics of brain activity and find it difficult to reveal the dynamic dependencies and causal interactions between brain regions, limiting their effectiveness in brain function research and clinical applications. To address this issue, we propose a **Spike-based Digital Brain** (Spike-DB), a novel fundamental model that introduces the spike computing paradigm into brain time series modeling. Spike-DB encodes fMRI signals as spike trains and learns the temporal driving relationships between anchor and target regions to achieve high-precision prediction of brain activity and reveal underlying causal dependencies and dynamic relationship characteristics. Based on Spike-DB, we further conducted downstream tasks including brain disease classification, abnormal brain region identification, and effective connectivity inference. Experimental results on real-world epilepsy datasets and the Alzheimer's Disease Neuroimaging Initiative (ADNI) dataset show that Spike-DB outperforms existing mainstream methods in both prediction accuracy and downstream tasks, demonstrating its broad potential in clinical applications and brain science research. Our code is available at `https://github.com/UAIBC-Brain/Spike-DB`.

## 1 Introduction

The human brain is one of the most complex dynamic systems, and its activity patterns underlie cognition, emotion, and behavior (Bassett & Gazzaniga, 2011). Therefore, in-depth analysis of brain activity is of central importance in neuroscience and clinical medicine (Poldrack & Farah, 2015). Functional magnetic resonance imaging (fMRI), currently the most widely used non-invasive imaging technique, can reflect the spatiotemporal dynamics of neural activity at a brain-wide scale by measuring temporal variations in the blood oxygenation level-dependent (BOLD) signal (Kim & Bandettini, 2023). However, fMRI signals typically have a low signal-to-noise ratio and are susceptible to interference from factors such as physiological noise and scanning conditions (Zhu et al., 2023; Misaki & Bodurka, 2021; Vizioli et al., 2021). This poses a significant challenge for extracting reliable and stable neural activity signatures from complex brain network data. To address this issue, brain activity analysis urgently requires high-precision modeling methods, multimodal information fusion, and advanced computational techniques (Tong & Pratte, 2012; Lee et al., 2024; Yuan et al., 2022).

Recently, various fundamental models have been proposed for analyzing brain activity, driven by advances in neuroscience and artificial intelligence. Early methods relied primarily on statistical modeling and signal decomposition techniques, such as independent component analysis (ICA) (Beckmann et al., 2005) and dynamic causal modeling (DCM) (Friston et al., 2003). Currently, deep learning (DL) methods have gained increasing attention. Frameworks such as graph neural network (GNN) (Nardi et al., 2024), Transformer (Bhatti et al., 2024), and contrastive learning (Zong et al.,

---

*Corresponding author.

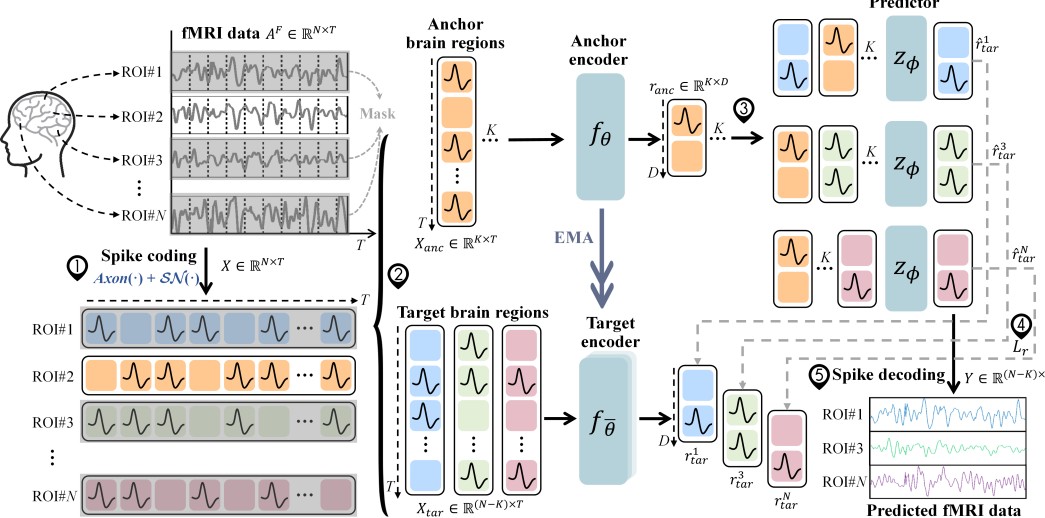

Figure 1: The proposed **Spike-based Digital Brain (Spike-DB)** architecture. (1) The input fMRI data is encoded as a spike train. (2) A subset of brain regions are randomly sampled as anchor regions, and the remaining regions are selected as target regions, and are fed to the anchor and target encoders, respectively. (3) The predictor takes as input the output of the anchor encoder and the mask token of each target region to be predicted, and predicts the representation of the target region conditioned on the position embedding. (4) The representation of the target region corresponds to the output of the target encoder, and its weights are updated at each iteration by the Exponential Moving Average (EMA) of the anchor encoder weights. (5) Spike decoding restores the embedding space features generated by the predictor to the original brain region space.

2024) are now widely used to capture the high-dimensional spatiotemporal dependencies between brain regions. Benefiting from the advances in fundamental models, the neuroscience community has begun exploring large-scale pre-training strategies. For example, models such as BrainLM (Caro et al., 2023), BrianMass (Yang et al., 2024), and Brain-JEPA (Dong et al., 2024) employ predictive or generative learning mechanisms to learn universal and robust representations from large-scale neuroimaging data, demonstrating excellent generalization capabilities in multi-task and cross-modal brain activity analysis.

Inspired by the above fundamental models, digital brain models have emerged, offering novel insights for modeling neural activities at the whole-brain scale. Concepts such as the virtual brain (Jirsa et al., 2023), digital twin brain (Lu et al., 2023), and virtual twin brain (Wang et al., 2024), as well as more advanced neural activity-based models (Wang et al., 2025), have been proposed to enable individualized brain digital mapping through the integration of multimodal imaging and neural dynamics modeling. However, these digital brain models are often built on statistical or continuous-valued neural network frameworks. Although they can capture macroscopic temporal patterns and functional connectivity (Beckmann et al., 2005), they struggle to reflect the discrete firing characteristics and biological constraints in brain activity (Gao & Wu, 2016), thus limiting their ability to interpret the mechanisms of real-world neural activity. Therefore, developing brain activity modeling frameworks that are more closely aligned with biological neuroscience has become an urgent need.

To address these challenges, we propose the Spike-based Digital Brain Model (Spike-DB), a novel fundamental model for brain activity analysis. This framework introduces the spike computing paradigm into brain time series modeling, enabling the digital brain to reproduce dynamic characteristics that are closer to the biological nervous system. Spike-DB can achieve high-precision prediction of brain activity by learning the temporal driving relationships of some brain regions to other brain regions, while revealing potential abnormal patterns and effective connectivity in brain diseases. The schematic diagram of our proposed method is shown in Figure 1. Specifically, we first encode the fMRI time series into a spike train, and randomly divide the anchor brain region and the target region in the brain region. By learning the temporal driving relationship of the anchor

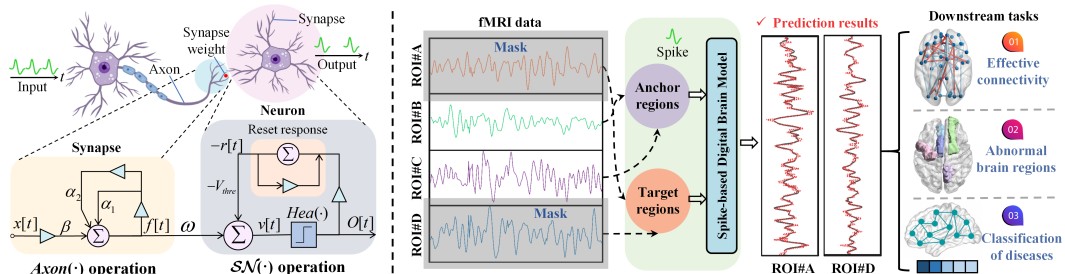

Figure 2: Left: The SNN model simulated by the IIR filter shows the connection structure composed of two biological neurons and presents the neural simulation process of the two operations $Axon(\cdot)$ and $\mathcal{SN}(\cdot)$. Right: Example of anchor and target masking strategy. Given four brain regions (A, B, C, D), we randomly sample two anchor regions (B, C) and use the remaining regions (A, D) as target regions. We use Spike-DB to predict fMRI data of regions A and D. The prediction results can be applied to three downstream tasks: effective connectivity (EC) quantification, abnormal brain regions detection, and brain disease classification.

region to the target region, we then obtain a dynamic prediction of the target region. Finally, the prediction results are restored to the original brain region space through spike decoding, achieving accurate modeling and prediction of brain activity. In this work, our contributions are summarized as follows:

- We propose a Spike-based Digital Brain (Spike-DB), which introduces the spike computing paradigm into brain time series modeling, thereby achieving high-precision prediction of brain activity.

- Based on Spike-DB, we further explore downstream tasks such as brain disease classification, abnormal brain region identification, and effective connectivity inference, demonstrating its broad potential in clinical applications and brain science research.

- We evaluate the performance of our proposed method using a real-world epilepsy dataset and the public Alzheimer's Disease Neuroimaging Initiative (ADNI) dataset, and the results show that the proposed method outperforms existing methods.

## 2 METHOD

### 2.1 SPIKING NEURON MODEL

The traditional Leaky Integrate-and-Fire (LIF) neuron (Gerstner & Kistler, 2002) mainly focuses on the dynamics of the membrane potential while ignoring the synaptic dynamic characteristics and the internal signal filtering mechanism of neurons. Therefore, it has limitations in characterizing the high- and low-frequency components and the long-term dependencies across brain regions that are reflected in the BOLD signals of fMRI data. To overcome these limitations, we introduce LIF neurons formulated with an Infinite Impulse Response (IIR) filter (Fang et al., 2020), leveraging their recursive structure to capture the temporal dynamic features of fMRI data.

We denote the input spike train as a sequence of time-shifted Dirac delta functions: $x_i = \sum_j \delta(t - t_i^j)$, where $t_i^j$ denotes the arrival time of the $j_{th}$ spike from the $i_{th}$ input synapse. Similarly, the output spike train can be defined as $O(t) = \sum \delta(t - t^f)$, $t^f \in \{t^f : v(t^f) = V_{thre}\}$. Thus, the SNN model is expressed as a network of IIR filters with nonlinear neurons:

$$v[t] = -V_{thre}r[t] + \sum_i^M \omega_i f_i[t] \tag{1}$$

$$r[t] = e^{\frac{-1}{\tau_r}} r[t-1] + O[t-1] \tag{2}$$

$$O[t] = Hea(v[t] - V_{thre}) \tag{3}$$

$$f_i[t] = \alpha_1 f_i[t-1] + \alpha_2 f_i[t-2] + \beta x_i[t-1] \tag{4}$$

where $t$ is the time, $v[t]$ is the neuron membrane potential, $V_{thre}$ is the threshold potential, $\omega_i$ is the associated weight of the $i_{th}$ synapse, $M$ is the total number of synapses, $r[t]$ is the reset filter, $\tau_r$ controls the decay speed of the reset impulse, $f_i[t]$ is the second-order IIR filter of the $i_{th}$ synapse, $\alpha_1 = e^{\frac{-1}{\tau_m}} + e^{\frac{-1}{\tau_s}}$, $\alpha_2 = -e^{-\frac{\tau_m + \tau_s}{\tau_m \tau_s}}$, $\beta = e^{\frac{-1}{\tau_m}} - e^{\frac{-1}{\tau_s}}$, $\tau_m$ and $\tau_s$ are time constants, and $Hea(\cdot)$ is the Heaviside step function, if $x \geq 0$, $Hea(x) = 1$, otherwise $Hea(x) = 0$.

Let $Axon(\cdot)$ represent the operation of transmitting a spike to the next neuron, which can be simulated by a second-order exponential IIR filter in Eq. 4. Let $\mathcal{SN}(\cdot)$ represent the behavior of the neuron updating the membrane potential and firing spikes, as shown in Eq. 1 to 3. Therefore, the SNN model simulated by the IIR filter can be divided into two operations, $Axon(\cdot)$ and $\mathcal{SN}(\cdot)$, as illustrated on the left in Figure 2.

## 2.2 Spike-based Digital Brain

We now introduce the proposed Spike-Based Digital Brain (Spike-DB), a non-generative framework for self-supervised learning in the space of spike representations, as illustrated in Figure 1. The core idea of Spike-DB is to randomly extract a subset of brain regions from the full fMRI time series as anchor regions and predict the fMRI time series of the remaining target regions. We use the Spike Transformer (Yao et al., 2023) as the underlying architecture for the anchor encoder, target encoder, and predictor. Operating in the spike space, the Spike Transformer effectively encodes fMRI time series into event-driven spike trains, preserving their dynamic information while reducing redundancy and noise (Ye et al., 2023).

**Spike encoding of fMRI time series.** We employ an IIR filter-emulated SNN model to capture temporal features in fMRI. Each subject's fMRI time series data is represented as $A^F \in \mathbb{R}^{N \times T}$, where $N$ denotes the number of brain regions and $T$ denotes the length of the time series. To this end, we can obtain the spike-based representation of the fMRI time series after $Axon(\cdot)$ and $\mathcal{SN}(\cdot)$ operations with the input $A^F$:

$$X_l = \mathcal{SN}_l(Axon_l(X_{l-1})), \quad l = 1, ..., L \tag{5}$$

where $l$ is the layer index, $X_0 = A^F$, and the final fMRI spike train is represented as $X \in \mathbb{R}^{N \times T}$.

**Anchor brain regions.** Spike-DB aims to predict the representations of $N - K$ target regions based on the representations of $K$ anchor regions. For an input fMRI spike train $X \in \mathbb{R}^{N \times T}$, anchor regions $X_{anc} = \{x_1, ..., x_i, ..., x_K\} \in \mathbb{R}^{K \times T}$ are obtained by randomly sampling brain regions in the range $\{1, 2, ..., N\}$. Next, $X_{anc}$ is fed into the anchor encoder $f_\theta$ to generate the corresponding region-level representation $r_{anc}$:

$$r_{anc} = \{r_{x_i}\}_{i \in \rho_{anc}} \tag{6}$$

where $r_{anc} \in \mathbb{R}^{K \times D}$, $D$ is the embedding dimension of each brain region, $\rho_{anc}$ represents the mask associated with the anchor brain region $X_{anc}$, $r_{x_i}$ is the representation of the $i$-th ($i = 1, 2, ..., K$) brain region, and the parameter $\theta$ is updated by gradient descent.

**Target brain regions.** In Spike-DB, the target brain region $X_{tar} \in \mathbb{R}^{(N-K) \times T}$ corresponds to the fMRI spike train representation obtained from the remaining $N - K$ brain regions, i.e., $X_{tar} = \{x_1, ..., x_j, ..., x_{N-K}\}$. We feed this into the target encoder $f_{\bar{\theta}}$ to obtain the corresponding brain region-level representation $r_{tar}^j$:

$$r_{tar}^j = \{r_{x_j}\}_{j \in \rho_{tar}^j} \tag{7}$$

where $r_{tar}^j \in \mathbb{R}^{1 \times D}$, $\rho_{tar}^j$ represents the mask associated with the target brain region $x_j$, and $r_{x_j}$ is the representation of the $j$-th ($j = 1, 2, ..., N - K$) brain region. In particular, the target brain region is obtained by masking the output representation of the target encoder rather than the input, and the parameter $\bar{\theta}$ is updated using an Exponential Moving Average (EMA) to smooth the training process and reduce noise, thereby obtaining a more stable representation (Assran et al., 2023).

**Prediction.** Given the output $r_{anc}$ of the anchor encoder, we aim to predict $N - K$ target region representations $r_{tar}^1, r_{tar}^2, ..., r_{tar}^{N-K}$. Therefore, for a given target region $r_{tar}^j$ and its corresponding target mask $\rho_{tar}^j$, the predictor $z_\phi(\cdot, \cdot)$ takes as the input $r_{anc}$ and the mask token $\{m_j\}_{j \in \rho_{tar}^j}$ for

each target region, and outputs each target region prediction $\hat{r}_{tar}^j$:

$$\hat{r}_{tar}^j = \{\hat{r}_{x_j}\}_{j \in \rho_{tar}^j} = z_\phi(r_{anc}, \{m_j\}_{j \in \rho_{tar}^j}) \tag{8}$$

where $\hat{r}_{tar}^j \in \mathbb{R}^{1 \times D}$ and the parameters $\phi$ are updated via gradient descent. The mask tokens are parameterized by a shared learnable vector and represented in conjunction with positional embeddings. Since predictions need to be made for each of the $N-K$ target regions, we apply the predictor $N-K$ times, each time conditioned on the mask tokens corresponding to the target brain region location, to obtain predictions $\hat{r}_{tar}^1, \hat{r}_{tar}^2, ..., \hat{r}_{tar}^{N-K}$ in sequence. To facilitate understanding, we provide a simple example of predicting the target brain region based on the anchor brain region and its applicable downstream tasks in the right part of Figure 2. Furthermore, during training, the training loss $L_r$ is defined as the mean squared error (MSE) between $r_{tar}^j$ and its corresponding prediction $\hat{r}_{tar}^j$ in terms of spike firing rates:

$$L_r = \frac{1}{N-K} \sum_{j=1}^{N-K} (\hat{R}_{tar}^j - R_{tar}^j)^2 \tag{9}$$

where $R_{tar}^j = \frac{1}{D} \sum_{d=1}^{D} r_{tar}^j$, $\hat{R}_{tar}^j = \frac{1}{D} \sum_{d=1}^{D} \hat{r}_{tar}^j$, which represent the average spike firing rates of $r_{tar}^j$ and $\hat{r}_{tar}^j$ in embedding dimension $D$ respectively.

**Spike Decoding.** To restore the embedding features generated by the predictor the original brain region space, we add a spike decoder at the end of the model. We first concatenate the predicted values $r_{tar}^1, r_{tar}^2, ..., r_{tar}^{N-K}$ of the target brain region according to the brain region index to form the overall target embedding matrix $\hat{r}_{tar}$:

$$\hat{r}_{tar} = concat(\hat{r}_{tar}^1, \hat{r}_{tar}^2, ..., \hat{r}_{tar}^{N-K}) \tag{10}$$

where $\hat{r}_{tar} \in \mathbb{R}^{(N-K) \times D}$. Essentially, our task is a regression task, where we reduce the spike train to the predicted fMRI time series $Y \in \mathbb{R}^{(N-K) \times T}$ by applying a fully connected layer, represented as $Y = Linear(\hat{r}_{tar})$.

## 3 EXPERIMENTS AND RESULTS

This section evaluates the effectiveness of our proposed Spike Digital Brain Model (Spike-DB) with extensive experiments. Our goal is to address the following research questions:

> **RQ1.** How does Spike-DB compare to various state-of-the-art models?
> **RQ2.** Can Spike-DB identify and explain abnormal brain regions or effective connectivity associated with neurological disorders such as epilepsy and AD, and whether its findings is consistent with clinical observations and existing medical research?
> **RQ3.** Do the Spike mechanism and brain region selection strategy play a significant role in the model's performance?

### 3.1 EXPERIMENT SETTINGS

**Implementation details.** In this study, we conduct experiments using the publicly available Alzheimer's Disease Neuroimaging Initiative (ADNI) dataset and a collected epilepsy dataset. The datasets are described in detail in the Appendix A.1. The epilepsy and ADNI datasets are divided into four disease datasets and two healthy datasets: FLE ($T = 240$), TLE ($T = 240$), SMC ($T = 197$), EMCI ($T = 197$), NC-E ($T = 240$), and NC-A ($T = 197$). For each dataset, 80% of the data is allocated for pre-training and 20% for testing prediction accuracy and downstream tasks (abnormal brain region detection and quantification of effective connectivity). In all tasks, the division of the training and test sets is based on the subjects rather than the data. Each dataset corresponds to a pre-trained Spike-DB model: $\delta_{FLE}, \delta_{TLE}, \delta_{SMC}, \delta_{EMCI}, \delta_{NC-E}$ and $\delta_{NC-A}$. For details on the implementation of the downstream brain disease classification task, see Appendix A.5. The pre-training parameter settings are shown in Appendix A.2. Notably, we fix the number of anchor regions $K$ to 89 in the pre-trained models above. This leaves only one target region, and

Table 1: Results of fMRI time series prediction task achieved by all methods on six data types (i.e., FLE, TLE, NC-E, EMCI, SMC, and NC-A), averaged over five independent runs. ↑: the higher, the better; ↓: the lower, the better. The best results are in **bold**, underlined indicate significant improvements over previous methods ($p < 0.05$).

| Method | Metric | FLE | TLE | NC-E | EMCI | SMC | NC-A |
|---|---|---|---|---|---|---|---|
| BrainLM | $R^2$ ↑ | $0.922_{\pm.043}$ | $0.948_{\pm.037}$ | $0.934_{\pm.048}$ | $0.907_{\pm.023}$ | $0.902_{\pm.043}$ | $0.936_{\pm.047}$ |
| | RSE ↓ | $0.279_{\pm.045}$ | $0.266_{\pm.054}$ | $0.258_{\pm.035}$ | $0.305_{\pm.027}$ | $0.313_{\pm.048}$ | $0.254_{\pm.029}$ |
| BrainMass | $R^2$ ↑ | $0.938_{\pm.044}$ | $0.953_{\pm.054}$ | $0.958_{\pm.044}$ | $0.930_{\pm.038}$ | $0.929_{\pm.049}$ | $0.957_{\pm.026}$ |
| | RSE ↓ | $0.249_{\pm.047}$ | $0.239_{\pm.029}$ | $0.204_{\pm.052}$ | $0.265_{\pm.022}$ | $0.267_{\pm.017}$ | $0.206_{\pm.042}$ |
| Brain-JEPA | $R^2$ ↑ | $0.952_{\pm.026}$ | $0.964_{\pm.026}$ | $0.967_{\pm.030}$ | $0.938_{\pm.039}$ | $0.943_{\pm.015}$ | $0.965_{\pm.023}$ |
| | RSE ↓ | $0.219_{\pm.016}$ | $0.217_{\pm.026}$ | $0.182_{\pm.057}$ | $0.249_{\pm.035}$ | $0.240_{\pm.019}$ | $0.188_{\pm.037}$ |
| BrainSymphony | $R^2$ ↑ | $0.963_{\pm.034}$ | $0.969_{\pm.040}$ | $0.974_{\pm.014}$ | $0.947_{\pm.034}$ | $0.948_{\pm.037}$ | $0.975_{\pm.026}$ |
| | RSE ↓ | $0.192_{\pm.043}$ | $0.202_{\pm.042}$ | $0.162_{\pm.055}$ | $0.231_{\pm.029}$ | $0.228_{\pm.040}$ | $0.160_{\pm.047}$ |
| **Spike-DB** | $R^2$ ↑ | $\underline{\mathbf{0.972}}_{\pm.017}$ | $\underline{\mathbf{0.979}}_{\pm.013}$ | $\underline{\mathbf{0.983}}_{\pm.016}$ | $\underline{\mathbf{0.954}}_{\pm.039}$ | $\underline{\mathbf{0.957}}_{\pm.041}$ | $\underline{\mathbf{0.981}}_{\pm.026}$ |
| | RSE ↓ | $\underline{\mathbf{0.168}}_{\pm.019}$ | $\underline{\mathbf{0.184}}_{\pm.012}$ | $\underline{\mathbf{0.131}}_{\pm.023}$ | $\underline{\mathbf{0.213}}_{\pm.029}$ | $\underline{\mathbf{0.207}}_{\pm.038}$ | $\underline{\mathbf{0.137}}_{\pm.026}$ |

the fMRI time series for all regions are predicted recurrently. Experimental results show that our model achieves optimal performance under this setting. Related ablation studies are described in Section 3.6.

**Metrics.** For the fMRI time series prediction task, we employ the root relative squared error (RSE) and the coefficient of determination ($R^2$). For the brain disease classification task, we employ the accuracy (ACC) and F1 score. The evaluation metrics are detailed in Appendix A.3.

## 3.2 MAIN RESULTS (**RQ1**)

We compared Spike-DB with a range of state-of-the-art models to evaluate its performance. The selected baselines cover a variety of architectural methods, from specialized deep learning models to large-scale fundamental models: 1) BrainLM (Caro et al., 2023): A large-scale fundamental model that learns fMRI representations using a masked autoencoding strategy for time series data. 2) Brain-Mass (Yang et al., 2024): A brain network-based model learned through large-scale self-supervised learning, using masked ROI modeling and representation alignment to pre-train a Transformer encoder. 3) Brain-JEPA (Dong et al., 2024): Leveraging the Joint Embedding Prediction Architecture (JEPA) to learn powerful representations by predicting masked data in a latent space. 4) BrainSymphony (Khajehnejad et al., 2025): A lightweight Transformer-based model that processes fMRI time series through parallel spatial and temporal streams and unifies their representation with perceptrons.

Table 1 reports the results of all methods on the fMRI time series prediction task. From Table 1, we can observe that our Spike-DB achieves state-of-the-art performance across all datasets. Compared to the best baseline model (BrainSymphony), our Spike-DB achieves comparable or even better performance. For example, on the epilepsy and ADNI datasets, BrainSymphony and Spike-DB achieve an average improvement of 1% and 0.8% on $R^2$, and 14.9% and 11.4% on RSE, respectively. Furthermore, on the relatively more challenging EMCI and SMC data, Spike-DB maintains state-of-the-art prediction performance. Spike-DB also achieves the highest $R^2$ and lowest RSE in the normal control groups NC-E and NC-A. These results demonstrate that spike-based digital brain modeling can better capture the temporal dependencies and nonlinear characteristics of brain regions.

## 3.3 CLASSIFICATION PERFORMANCE (**RQ1**)

To comprehensively evaluate the effectiveness of Spike-DB in brain disease classification, we further introduced models designed specifically for brain disease classification in recent years: Starformer (Dong et al., 2025) and STCAL (Liu et al., 2023), as comparison methods. Table 2 summarizes the results of different methods in four brain disease classification tasks: FLE vs. NC-E, TLE vs. NC-E, EMCI vs. NC-A, and SMC vs. NC-A. It is not difficult to see that BrainLM, STCAL, BrainMass, and Brain-JEPA perform well overall, but lag behind BrainSymphony and Starformer in terms of

Table 2: Results of brain disease classification achieved by all methods on four binary classification tasks (i.e., FLE vs. NC-E, TLE vs. NC-E, EMCI vs. NC-A, and SMC vs. NC-A), averaged over five independent runs. The best results are in **bold**, underlined indicate significant improvements over previous methods ($p < 0.05$).

| Method | FLE vs. NC-E | | TLE vs. NC-E | | EMCI vs. NC-A | | SMC vs. NC-A | |
|---|---|---|---|---|---|---|---|---|
| | ACC(%) | F1(%) | ACC(%) | F1(%) | ACC(%) | F1(%) | ACC(%) | F1(%) |
| BrainLM | 84.26±3.96 | 82.11±2.61 | 83.17±4.19 | 77.92±4.20 | 78.79±4.35 | 70.83±3.20 | 80.56±3.65 | 75.29±2.69 |
| STCAL | 86.11±2.38 | 84.21±4.13 | 85.15±4.27 | 81.93±2.64 | 81.82±2.09 | 76.00±4.48 | 81.48±3.01 | 75.61±2.03 |
| BrainMass | 87.04±2.79 | 87.76±3.12 | 86.14±2.76 | 82.50±2.94 | 83.33±3.44 | 78.43±3.07 | 81.69±3.89 | 82.35±3.04 |
| Brain-JEPA | 87.96±2.72 | 86.59±3.48 | 87.13±3.01 | 83.95±4.04 | 84.85±2.71 | 81.48±1.18 | 83.10±4.25 | 81.82±3.28 |
| BrainSpmphony | 88.89±2.24 | 87.76±3.82 | 88.12±3.32 | 85.37±3.55 | 85.88±3.45 | **86.21**±3.32 | 84.51±3.21 | 83.58±4.77 |
| Starformer | 89.81±3.16 | 88.89±3.06 | 89.11±3.54 | 86.75±3.58 | 86.36±1.89 | 83.20±3.88 | 85.92±2.63 | 85.71±2.34 |
| **Spike-DB** | **91.67**±2.74 | **91.90**±2.37 | **90.10**±2.94 | **88.09**±2.70 | **87.88**±2.11 | 85.19±3.01 | **86.95**±2.64 | **87.32**±1.26 |

ACC and F1 on multiple tasks. Starformer achieves a high ACC on the FLE vs. NC-E and TLE vs. NC-E tasks, while BrainSymphony performs exceptionally well on the EMCI vs. NC-A task. However, Spike-DB achieves the best performance overall on all four tasks, achieving 86.95% ACC and 87.32% F1 on the more challenging SMC vs. NC-A task. These results further validate the effectiveness and robustness of Spike-DB in brain disease classification tasks.

### 3.4 MAPPING OF ABNORMAL BRAIN REGIONS DRIVEN BY SPIKE-DB (**RQ2**)

In this experiment, we apply the trained Spike-DB model of the NC group ($\delta_{NC-E}$, $\delta_{NC-A}$) to the data of FLE, TLE, EMCI, and SMC patients to predict their fMRI signals under normal constraints. For the patient's predicted results and original fMRI data, we perform group averaging processing in the dimension of the brain region. That is, for each sample, we calculate its mean fMRI signal within each brain region to obtain a group-level average representation $N \times T$. Subsequently, we compared and analyzed the original average fMRI data of the patient group with the predicted data of the model. To further identify abnormal brain regions, we perform statistical tests on the differences between the original and predicted data after group averaging. Specifically, we use paired $t$-tests to assess the significance of the differences between the two in each brain region. This method allows us to screen brain regions that show significant differences in the patient population compared to the model predictions, thereby inferring potential pathological abnormalities.

Statistical test results show that for the four disease data sets (FLE, TLE, EMCI, and SMC), a small number of brain regions exhibit significant differences ($p < 0.05$) between the original and predicted data. Specifically, we observe significant differences in four brain regions in the FLE data, three in the TLE data, five in the EMCI data, and two in the SMC data. We plot violin plots for the significantly abnormal brain regions to illustrate the distribution differences between the original data and the predicted results. These abnormal brain regions and their corresponding violin plots appear in Figure 3. The results indicate that significant deviations between the predicted and true signals exist in regions such as HIP, THA, and PCG, suggesting possible pathological abnormalities in these regions.

Notably, some of the identified brain regions are highly consistent with previous studies on epilepsy and ADNI. For example, studies have shown that patients with FLE often exhibit abnormal neural activity in the SMA region (Hong et al., 2023) and significantly increased functional connectivity in the HIP region (Englot et al., 2016). The pathogenesis of most TLE cases is closely linked to the HIP region (Tatum IV, 2012), and TLE patients exhibit higher functional connectivity in the SPG region than those with NC (Li et al., 2023). The PHG region is the first brain region to show structural and functional abnormalities in patients with EMCI and is directly associated with memory decline (van de Mortel et al., 2021). Reduced functional connectivity in the PCG region is one of the earliest indicators of abnormality in EMCI (He et al., 2021), and EMCI patients also show significantly reduced functional connectivity in the PCUN region (Xu et al., 2021). The HIP region is also one of the first brain regions to atrophy in patients with SMC (Smith et al., 2012), and the pathological changes of this disease are often accompanied by a decrease in bilateral THA volume. In summary, the abnormal brain regions identified by our Spike-DB model are not only statistically

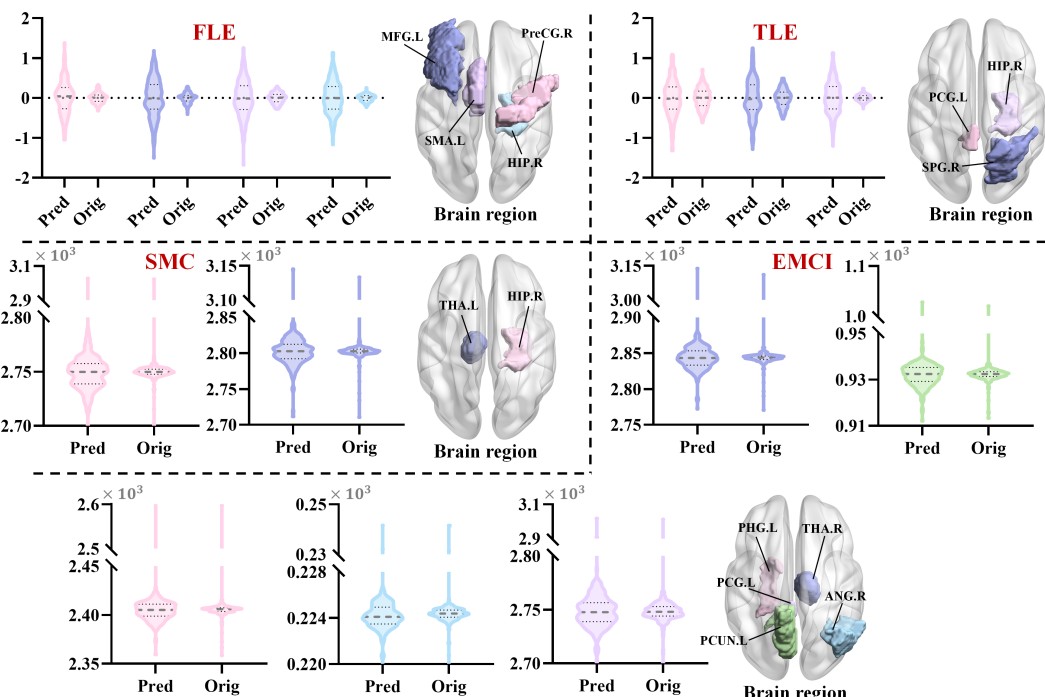

Figure 3: Visual distribution results of significantly abnormal brain regions in FLE, TLE, EMCI, and SMC patient data. Each brain region corresponds to a set of violin plots, showing the distribution differences between the patient's original data and the model-predicted data in these brain regions, and these differences are statistically significant ($p < 0.05$).

significant but also well-supported by existing clinical and imaging evidence, further validating the model's reliability in identifying abnormal brain regions.

## 3.5 PERTURBING THE TRAINED SPIKE-DB TO INFER EFFECTIVE CONNECTIVITY (**RQ2**)

### 3.5.1 GAUSSIAN PULSE-BASED PERTURBATION STRATEGY

After training, we perturb each input region of Spike-DB to infer global effective connectivity (EC) (Luo et al., 2025). Perturbation involves selectively increasing the signal of a specific region at time $[0, T]$ while leaving other regions unperturbed. The EC from region $i$ to all other regions is quantified as the difference in model predictions between the perturbed and unperturbed conditions:

$$EC_i = \delta(X + \Delta \cdot e_i) - \delta(X) \tag{11}$$

where $e_i$ is a unit vector, with a value of 1 at the $i$-th brain region and 0 elsewhere, representing the perturbation in the $i$-th region. $EC_i > 0$ represents excitatory connectivity, and the opposite represents inhibitory connectivity. $\Delta$ represents the perturbation strength. Because Spike-DB operates on the spike space, conventional perturbation methods are difficult to apply directly. To this end, we designed a perturbation strategy based on Gaussian pulses (Breakspear, 2017). This introduces a local, smooth, and amplitude-controlled Gaussian pulse signal into the fMRI spike time series to simulate the impact of instantaneous perturbations on brain activity. Its mathematical form is:

$$\Delta = k \cdot \exp(-\frac{(T - t_0)^2}{2\alpha^2}) \tag{12}$$

where $k$ is the disturbance amplitude (set to 1), $t_0$ is the pulse center time (set to $\frac{T}{2}$), and $\alpha$ controls the pulse width (set to 20).

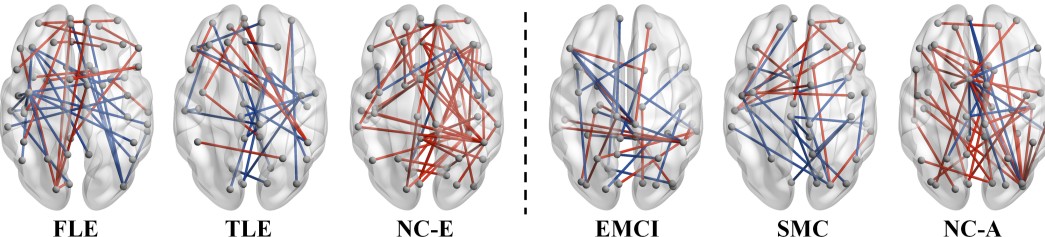

**FLE**  **TLE**  **NC-E**   **EMCI**  **SMC**  **NC-A**

Figure 4: Visualization of the ECs inferred from six pre-trained models ($\delta_{FLE}$, $\delta_{TLE}$, $\delta_{SMC}$, $\delta_{EMCI}$, $\delta_{NC-E}$, and $\delta_{NC-A}$), with one representative subject randomly selected from each dataset. Red lines indicate excitatory connections, and blue lines indicate inhibitory connections.

### 3.5.2 PERTURBATION-DRIVEN EC ANALYSIS

In this experiment, we visualize the ECs inferred from various data types using six pre-trained models ($\delta_{FLE}$, $\delta_{TLE}$, $\delta_{SMC}$, $\delta_{EMCI}$, $\delta_{NC-E}$, and $\delta_{NC-A}$). Figure 4 displays the ECs of one representative subject randomly selected from each data type. Observations are as follows.

First, in both the epilepsy and ADNI datasets, the patient groups (FLE, TLE, EMCI, and SMC) exhibit significantly fewer ECs than their corresponding normal controls (NC-E and NC-A). Second, in the normal controls, the number of excitatory connections significantly outnumbered inhibitory connections, whereas in the patient groups, the reverse pattern was observed. We speculate that these differences may reflect significant changes in the functional connectivity patterns of brain networks, particularly an imbalance in the ratio of excitatory to inhibitory connections. Notably, this imbalance may indicate a decrease in the efficiency and stability of neural network information processing. Related research supports this view. For example, McCormick & Contreras (2001) suggested that an imbalance between excitatory and inhibitory influences may cause epileptic seizures. Using multiple learning algorithms, Ji et al. (2021) and Huang et al. (2022) quantify ECs and show that the number of connections in the NC group is significantly greater than that in the Alzheimer's disease (AD) group. These experimental results demonstrate that our Spike-DB can not only identify effective connectivity patterns, but also further verify the close relationship between the imbalance of excitation-inhibition balance in brain connections and the occurrence and development of diseases.

### 3.6 ABLATION STUDY (**RQ3**)

To validate the effectiveness of the key components of the proposed Spike-DB model, we design and conduct two sets of ablation experiments on the six data types. Specifically, we remove the spiking neuron layer from all components, which not only eliminates the spiking coding mechanism but also degenerates the Spike Transformer into a regular Transformer. The results are shown in Figures 5. First, we compare the model's performance with and without the spike mechanism. The results show that removing the spike mechanism significantly degrades the prediction performance of Spike-DB in both R² and RSE. This demonstrates that spike representation plays a key role in capturing fMRI temporal dependencies (Lv et al., 2024) and dynamic brain activity patterns (Shibo et al., 2025), and is essential for the model to learn functional brain features effectively.

Next, we gradually reduce the number of anchor regions $K$ to assess its impact on model performance. Experimental results show that as $K$ decreases, overall model performance declines. In particular, when the number is less than half (i.e., $K < 45$), model performance deteriorates significantly, with a substantial decrease in R² and a notable increase in RSE. It is worth noting that when $K = 70$, the average value of R² is 0.949, and the average value of RSE is 0.214. A comparison with Table 1 shows that Spike-DB still outperforms BrainLM and BrainMass. This result suggests that an appropriate number of anchor regions is crucial for effectively constraining the model's learning process, and too few anchor regions weaken the model's ability to capture interactions between brain regions.

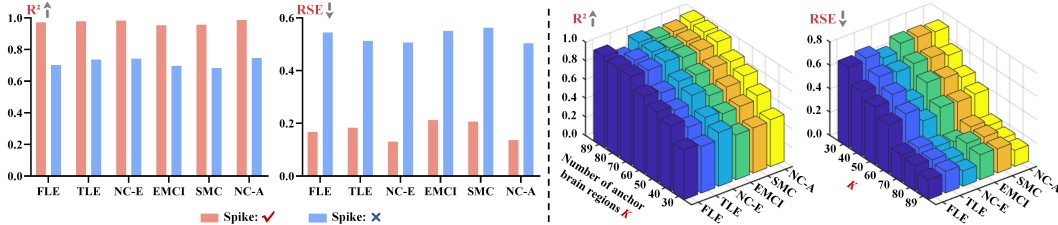

Figure 5: Left: Comparison of model performance when retaining the spike mechanism ($\sqrt{}$) and removing the spike mechanism ($\times$). Right: Effect of different numbers of anchor brain regions $K$ on model performance.

## 4 CONCLUSION

In this work, we develop the Spike-Based Digital Brain (Spike-DB), a novel fundamental model for brain activity analysis. Spike-DB introduces the spike computing paradigm to brain time series modeling to simulate the dynamic characteristics of biological neural systems. This model not only enables highly accurate predictions of brain activity but also assists in brain disease classification, identifying abnormal brain regions, and inferring effective connectivity. Through extensive experiments on epilepsy and ADNI datasets, we validate Spike-DB's advantages over existing methods and demonstrate its excellent generalization across a variety of tasks. Our method promises that spike-based digital brain modeling will open new avenues for clinical diagnosis and brain science research.

### ETHICS STATEMENT.

The experiments on the epilepsy dataset are approved by the Ethics Committee of Nanjing University Jinling Hospital. All experiments are conducted in accordance with relevant guidelines and regulations. All participants provide written informed consent before participation.

### ACKNOWLEDGMENTS

This study was supported in part by the National Natural Science Foundation of China (No. 62471259, 62472228, 62376123), and the Natural Science Foundation of Jiangsu Higher Education Institutions of China (No. 24KJB520032).

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

# A APPENDIX

## A.1 SUBJECTS AND DATA PREPROCESSING

In this study, we conduct experiments using the publicly available Alzheimer's Disease Neuroimaging Initiative (ADNI) dataset[1] and a collected epilepsy dataset. The two datasets are described below: ADNI dataset: We download fMRI data of 203 subjects from ADNI 2 and ADNI 3 datasets, comprising 73 normal controls (NC), 70 patients with significant memory concern (SMC), and 60 patients with early mild cognitive impairment (EMCI). Epilepsy dataset: This dataset contains 103 patients with frontal lobe epilepsy (FLE), 89 patients with temporal lobe epilepsy (TLE), and 114 NCs. All subjects are recruited at Jinling Hospital, Nanjing University Medical School. Data are collected using a Siemens Trio 3 T MRI scanner. fMRI scans are acquired using a single-shot, gradient-echo planar imaging sequence. Scan parameters are as follows: repetition time $= 2000\ ms$, echo time $= 30\ ms$, flip angle $= 90°$, 30 transverse slices, field of view (FOV) $= 240 \times 240\ mm^2$, slice thickness $= 4\ mm$, interstice gap $= 0.4\ mm$, voxel size $= 3.75 \times 3.75 \times 3.75\ mm^3$.

We use SPM12 version 2.0 of the DPARSF toolkit to preprocess all fMRI data in the ADNI and epilepsy datasets. First, the initial functional time series are slice-time acquisition corrected, re-aligned, and normalized to the echo-planar imaging (EPI) template. Subsequently, detrending is performed to eliminate sources of spurious variance (i.e., six head motion parameters, averaged signals from cerebrospinal fluid and white matter, and global brain signal), followed by bandpass filtering (0.01 to 0.08 Hz) on the time series data. Finally, the data are divided into 90 regions of interest (ROIs) based on the automatic anatomical Labeling (AAL) template. The functional data for each subject in the epilepsy dataset contain 240 time points, and the functional data for each subject in the ADNI dataset contain 197 time points.

## A.2 PARAMETER SETTINGS

The proposed model and algorithm are implemented based on PyTorch and snnTorch (Eshraghian et al., 2023), and trained on a single GPU (NVIDIA RTX 4090 with 24GB memory). The pre-training parameter settings are shown in Table 3.

Table 3: Pre-training settings.

| Config | Value |
|---|---|
| training epoch | 100 |
| threshold potential $V_{\text{thre}}$ | 1 |
| decay speed $\tau_r$ | 0.9 |
| time constant $\tau_m$ | 4 |
| time constant $\tau_s$ | 1 |
| dropout | 0.1 |
| embedding dimension $D$ | 64 |
| multi-head attention | 8 |
| encoder layer | 6 |
| predictor layer | 4 |
| layer index $L$ | 3 |
| optimizer | AdamW |
| optimizer momentum | $\beta_1 = 0.9$ 
 $\beta_2 = 0.999$ |
| learning rate | $1 \times 10^{-4}$ |
| EMA schedule | linear |
| EMA start momentum | 0.996 |
| EMA final momentum | 1 |

---

[1]https://adni.loni.usc.edu/

Table 4: Results of fMRI time series prediction task achieved by all methods on the additional AMRD dataset (i.e., MD and NC-M data types), averaged over five independent runs. ↑: the higher, the better; ↓: the lower, the better. The best results are in **bold**, underlined indicate significant improvements over previous methods ($p < 0.05$).

| Method | Metric | MD | NC-M |
|---|---|---|---|
| BrainLM | R² ↑ | $0.936_{\pm.041}$ | $0.931_{\pm.037}$ |
| | RSE ↓ | $0.253_{\pm.027}$ | $0.262_{\pm.034}$ |
| BrainMass | R² ↑ | $0.945_{\pm.037}$ | $0.946_{\pm.036}$ |
| | RSE ↓ | $0.234_{\pm.049}$ | $0.232_{\pm.052}$ |
| Brain-JEPA | R² ↑ | $0.951_{\pm.038}$ | $0.959_{\pm.026}$ |
| | RSE ↓ | $0.222_{\pm.012}$ | $0.203_{\pm.027}$ |
| BrainSymphony | R² ↑ | $0.957_{\pm.036}$ | $0.966_{\pm.017}$ |
| | RSE ↓ | $0.205_{\pm.035}$ | $0.183_{\pm.056}$ |
| Spike-DB | R² ↑ | $\mathbf{0.964}_{\pm.016}$ | $\mathbf{0.973}_{\pm.017}$ |
| | RSE ↓ | $\mathbf{0.190}_{\pm.030}$ | $\mathbf{0.164}_{\pm.010}$ |

## A.3 EVALUATION METRICS

For the evaluation of the fMRI time series prediction task, we use the relative squared error (RSE) and the coefficient of determination (R²), calculated as shown in Eq. 13 and 14. For the brain disease classification task, we report the accuracy (ACC = $\frac{TP+TN}{TP+TN+FP+FN}$) and F1 score (F1 = $\frac{2TP}{2TP+FP+FN}$), where FP, TP, FN and TN represent false positive, true positive, false negative, and true negative values, respectively.

$$RSE = \sqrt{\frac{\sum_{m=1}^{M} ||Y^m - \hat{Y}^m||^2}{\sum_{m=1}^{M} ||Y^m - \bar{Y}||^2}} \tag{13}$$

$$R^2 = \frac{1}{MCL} \sum_{m=1}^{M} \sum_{c=1}^{C} \sum_{l=1}^{L} \left[ 1 - \frac{\left(Y_{c,l}^m - \hat{Y}_{c,l}^m\right)^2}{\left(Y_{c,l}^m - \bar{Y}_{c,l}\right)^2} \right] \tag{14}$$

In these equations, $M$ represents the number of samples in the test set, $C$ is the number of channels, and $L$ is the prediction length. The true values for the $m$-th sample are denoted by $Y^m$, and their average over all samples is $\bar{Y}$. Specifically, $Y_{c,l}^m$ denotes the $l$-th future value of the $c$-th variable for the $m$-th sample, $\bar{Y}_{c,l}$ is the average of $Y_{c,l}^m$ across all samples. The predicted values corresponding to these true values are denoted by $\hat{Y}^m$ and $\hat{Y}_{c,l}^m$, respectively.

Compared with the mean square error (MSE) and mean absolute error (MAE), RSE and R² are more robust to outliers or values with large absolute values in the dataset, and are therefore widely used in time series forecasting tasks.

## A.4 RESULTS OF ADDITIONAL DATASETS

To further verify the robustness of Spike-DB, we expand the experiment by incorporating a larger dataset. This dataset is sourced from the Decoding Neurofeedback Project organized by the Agency for Medical Research and Development (AMRD) in Japan, encompassing 3T MRI data (resting-state fMRI EPI images, corrupted T1-weighted images, and field maps) from 1410 subjects (620 patients and 790 normal controls). We extract resting-state fMRI data from this dataset, covering 230 time points, and uniformly categorize the subjects into disease-related and disease-free groups. Ultimately, the dataset includes 849 subjects, of whom 384 are multi-disease (MD) patients and 465 are normal controls (NC-M). Data preprocessing is consistent with the epilepsy and ADNI datasets.

Table 4 reports the results of Spike-DB on the AMRD dataset for fMRI time series prediction. As shown in Table 4, Spike-DB maintains strong predictive power on large-scale datasets, with R² values of 0.964 and 0.973, and a significantly lower RSE value than other methods. Secondly, Table 5

Table 5: Results of brain disease classification achieved by all methods on the additional AMRD dataset (i.e., MD vs. NC-M), averaged over five independent runs. The best results are in **bold**, underlined indicate significant improvements over previous methods ($p < 0.05$).

| Method | MD vs. NC-M | |
|---|---|---|
| | ACC(%) | F1(%) |
| BrainLM | $74.69_{\pm 3.39}$ | $73.89_{\pm 2.84}$ |
| STCAL | $77.78_{\pm 3.40}$ | $72.73_{\pm 1.97}$ |
| BrainMass | $80.86_{\pm 4.94}$ | $74.14_{\pm 2.99}$ |
| Brain-JEPA | $82.08_{\pm 2.62}$ | $77.37_{\pm 2.06}$ |
| BrainSpmphony | $83.33_{\pm 1.41}$ | $78.05_{\pm 4.17}$ |
| Starformer | $83.82_{\pm 2.59}$ | $81.08_{\pm 3.09}$ |
| **Spike-DB** | $\underline{84.09_{\pm 2.64}}$ | $\underline{\mathbf{85.12_{\pm 2.89}}}$ |

reports the results of Spike-DB on the AMRD dataset for brain disease classification. Specifically, Spike-DB achieved an accuracy of 84.09% and an F1 score of 85.12%, both superior to other methods. This indicates that Spike-DB not only performs excellently in fMRI time series prediction but also achieves a leading position in brain disease classification, demonstrating its superior ability and consistency in handling large-scale datasets. This further validates the broad applicability and efficiency of Spike-DB in multiple brain disease tasks.

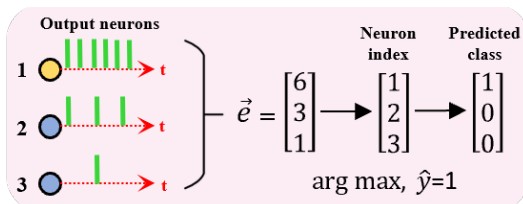

Figure 6: Classification method in spike space.

A.5 DETAILS OF DOWNSTREAM CLASSIFICATION TASKS

Unlike the main fMRI prediction task, which divides data into six data types, the training data used in the downstream task mainly includes four groups: FLE vs. NC-E, TLE vs. NC-E, EMCI vs. NC-A, and SMC vs. NC-A. To transition from a regression task to a classification task, we modified the model architecture, using the same encoder part and adding a classification head on top of it. Specifically, fMRI data is first encoded into spike trains using spiking neuron layer, then pre-trained using an encoder (anchor, target, and predictor), and used as a feature extractor for the downstream task. Finally, these extracted spike features are input into the classifier for brain disease classification.

Regarding the choice of classifier, since the main framework is in the spike space, traditional classifiers are no longer applicable. We will describe a vectorized implementation method for determining the predicted class based on the output spike sequence. Specifically, let $\vec{Z}[t] \in \mathbb{R}^m$ represent the spike fired by each output neuron over time, where $m$ is the number of output classes. Let $\vec{e} \in \mathbb{R}^m$ be the spike count for each output neuron, which can be obtained by summing $\vec{Z}[t]$ over $T$ time steps:

$$\vec{e} = \sum_{t=0}^{T} \vec{Z}[t] \tag{15}$$

the index of $\vec{e}$ with the maximum count corresponds to the predicted class $\hat{y}$:

$$\hat{y} = \arg\max_i e_i \tag{16}$$

To facilitate understanding, we provide an example in Figure 6: Neuron 1 fires 6 spikes, Neuron 2 fires 3 spikes, and Neuron 3 fires 1 spike. Compared to Neurons 2 and 3, Neuron 1 fires the most spikes, corresponding to a predicted class of 1. For the binary classification task, we used the classification cross-entropy loss function $L_c$:

$$L_c = -(\hat{y} \log \hat{y} + (1 - \hat{y}) \log(1 - y)) \tag{17}$$

where $y$ is the true class.

### A.6 Model complexity comparison

As shown in Table 6, the Spike-DB method demonstrates significant advantages in terms of parameter count and training time by introducing a spiking mechanism. Compared to other methods, Spike-DB has only 2.7 M parameters, far lower than Brain-JEPA (22.0 M) and BrainMass (14.4 M), while the training time per epoch is only 96.73 s, the shortest of all methods. This is thanks to the introduction of the spiking mechanism, which effectively reduces computational complexity and memory consumption by simulating the spiking firing pattern of neurons, allowing Spike-DB to significantly improve training efficiency while maintaining performance.

Table 6: The total number of parameters for all methods and their average training time per epoch (based on the average of six data types: FLE, TLE, NC-E, EMCI, SMC, and NC-A).

| Method | Parameters (M) | Training time (s) |
|---|---|---|
| BrainLM | 13.0 | 163.59 |
| BrainMass | 14.4 | 284.65 |
| Brain-JEPA | 22.0 | 127.81 |
| BrainSymphony | 2.0 | 114.18 |
| Spike-DB | 2.7 | 96.73 |

