# OpenReview forum: "Spike-based Digital Brain: a novel fundamental model for brain activity analysis"
_ICLR.cc/2026/Conference — ICLR 2026 Poster_

### Official Review · Reviewer_dTth · 2025-10-19

**Soundness:** 2
**Presentation:** 3
**Contribution:** 2
**Rating:** 4
**Confidence:** 3

**Summary:**

This paper proposes **Spike-DB**, a spike-based computing framework for modeling fMRI data. The model encodes fMRI signals into spike trains through a differentiable integrate-and-fire mechanism, applies a masked region prediction strategy to learn inter-regional dependencies, and decodes the resulting spike representations for reconstruction and analysis. Spike-DB achieves better performance than the compared state-of-the-art methods on data prediction and brain disease classification tasks. It also provides interpretability analyses, including abnormal region detection and effective connectivity mapping. The framework represents an attempt to unify spiking neural computation with brain modeling. However, its biological justification for applying spike encoding to slow fMRI signals is limited, and its evaluation on only two datasets constrains generalization.

**Strengths:**

1. While more biological justification is needed, the idea of using SNN-like processing for brain activity modeling represents an attempt to bridge computational neuroscience and machine learning.

2. The proposed Spike-DB framework is clearly structured and includes explicit mathematical formulations, making it easy to follow.

3. The manuscript is well organized and features high-quality figures.

4. The proposed Spike-DB model has been tested on diverse tasks.

**Weaknesses:**

1. The paper’s core claim is to “encode fMRI signals as spike trains.” However, fMRI BOLD signals are slow hemodynamic measures (around 1 Hz), not neural spikes (around 10-1000 Hz). Compared with fMRI, which has low temporal resolution, spike-based computing is more suitable for modalities with high temporal resolution, such as iEEG or EEG. How do the authors justify the biological meaning of such a design when applied to fMRI data?

2. The model uses generic spike encoders and decoders without linking parameters (τ_m, τ_s, thresholds) to realistic neural biophysics or cortical circuitry. Claims of “biological interpretability” are therefore not well supported.

3. The authors claim that they are the first to introduce the spike computing paradigm into brain time-series modeling. However, earlier works have treated fMRI data as spike trains, though in slightly different ways, for example:
   [a] Kasabov, Nikola K., Maryam Gholami Doborjeh, and Zohreh Gholami Doborjeh. *“Mapping, learning, visualization, classification, and understanding of fMRI data in the NeuCube evolving spatiotemporal data machine of spiking neural networks.”* IEEE Transactions on Neural Networks and Learning Systems 28.4 (2016): 887–899.
   Hence, this claim should be softened.

4. The authors fix the number of anchor regions to 89 out of 90 total, leaving only one region to predict. This setup trivializes the prediction task, as each region is almost self-predictive given its high autocorrelation.

5. Minor differences (about 1-2%) are presented as “state of the art,” which is not statistically justified. Standard deviations should also be reported to assess robustness across random seeds.

6. The paper reports disease classification results based on the learned spike representations, but the implementation details of this task are not clearly described. It is unclear whether a multi-layer perceptron, a linear classifier, or another model was used, and whether the SNN backbone was frozen or fine-tuned during training. The paper also omits details such as the loss function, optimization setup, and evaluation protocol. Clarifying these aspects and reporting variance across runs would strengthen the reproducibility and reliability of the classification results.

7. Only one public dataset and one collected dataset are used for evaluation, making it difficult to demonstrate the method’s generalization and reliability. More publicly available datasets should be included to evaluate the proposed approach.

8. The authors replace causal modeling with simple perturbation differences. Whether these results correlate with true causal effects needs to be carefully justified.

9. Finally, no code, model weights, or preprocessing scripts are provided, which harms reproducibility.

**Questions:**

1.How do you justify applying spike-based encoding to slow fMRI BOLD signals ( around 1 Hz)? Why is this paradigm appropriate for fMRI rather than higher-temporal-resolution modalities like EEG or iEEG?

2.How were τ_m, τ_s, and threshold values chosen? Do they have any neurophysiological grounding or sensitivity analysis?

3.How does Spike-DB differ fundamentally from earlier fMRI–SNN works (e.g., NeuCube, Kasabov et al., 2016)?

4.Will fix 89 anchor regions and predict only one be affected by fMRI’ high autocorrelation of near region?

5.Please report standard deviations and significance tests to confirm the 1–2% improvements.

6.What classifier (MLP, linear, etc.) was used? Was the backbone frozen or fine-tuned? What loss and evaluation protocol were applied?

7.Please evaluate Spike-DB on additional public datasets to demonstrate robustness and effectiveness.

8.How does your perturbation-based EC relate to established causal models (e.g., DCM, Granger)? Any validation against known connections?

9.Will you release code, model weights, and preprocessing scripts to enable replication?

I will raise my score if these issues are well addressed.

**Details Of Ethics Concerns:**

N.A.

---

> ### Author Response · Authors · 2025-11-15
> **Response regarding the biological rationale of this Paper**
>
> We thank the four reviewers for their insightful comments on this paper. We note that all four reviewers pointed out the infeasibility of encoding fMRI signals into spike trains. Therefore, we are addressing this issue first. Regarding other related experiments and questions, we will respond as soon as possible before the deadline and will do our best to meet the reviewers' expectations.
>
> First, our method does not directly convert fMRI signals into high-frequency spike trains at the neuron level; instead, it converts brain region-level fMRI signals into spike representations. These spike representations do not capture neuronal-level firing behavior, but rather efficiently encode brain region activity through spike trains. To avoid misunderstanding, we have revised the relevant content in the original paper.
>
> Our motivation for converting fMRI signals into spike representations stems from the classic hemodynamic response function (HRF), a mathematical model describing the dynamic response of cerebral blood flow to neural activity. Specifically, the core of the HRF model lies in its description of the impulse response of blood flow after stimulation; that is, after activation of a brain region, the blood flow response does not simply increase linearly, but exhibits a fluctuating pattern, initially rising and then gradually declining. This response pattern is similar to the spike firing process of neurons, where neurons rapidly fire in the form of spikes upon receiving stimulation. Although the timescale of the blood flow response is relatively long, it still captures the spike-like influence of neural activity on blood flow. In this way, we can draw an analogy between changes in fMRI signals and the spike firing of neurons, and use SNNs to simulate this temporal response. According to [Ref 1], the BOLD spike response $y(i,t)$ of fMRI signals can be represented as the convolution of the activity-induced signal and the HRF:
>
> $y(i,t)=\sum{c(i)\delta (t)\cdot h(t)+\sum{\beta (i)n(t)+\mu (i,t)}}$
>
> where $t$ is time, $c(i)$ is the amplitude of neural activity on the  $i$-th voxel, $h(t)$ is the HRF function, $\delta (t)$ is the Dirac-delta function, $n(t)$ is the known perturbation regression (movement or low-frequency drift, etc.), $\beta (i)$ is the correlation weight, and $\mu (i,t)$ is independently distributed Gaussian noise.
>
> Current research has converted fMRI signals into spike representations for further analysis. For example, Deco et al. [Ref 2] proposed the Brain Songs framework to explore the relevant time scales of human brain signals, demonstrating the effectiveness of encoding fMRI signals into spike trains in capturing brain functional connectivity. Previous studies [Ref 3-5] have shown a linear relationship between the BOLD signal of fMRI and the neuronal firing rate. Compared with these studies, our innovation lies in using SNNs to generate spike representations more naturally, rather than first inferring the BOLD signal as a continuous variable through a model and then converting it into pseudo-spike events, as is done in traditional methods.
>
> It is important to note that capturing neuronal-level firing behavior is indeed a very complex and challenging task, which cannot be accomplished solely with non-invasive devices such as fMRI and EEG. We anticipate that with advancements in data acquisition technologies, our models will be able to truly characterize the fine-grained activity at the neuronal level in the brain.
>
> [Ref 1] Karahanoğlu F I, Caballero-Gaudes C, Lazeyras F, et al. Total activation: fMRI deconvolution through spatio-temporal regularization[J]. Neuroimage, 2013, 73: 121-134.
>
> [Ref 2] Deco, Gustavo, Josephine Cruzat, and Morten L. Kringelbach. "Brain songs framework used for discovering the relevant timescale of the human brain." Nature communications 10.1 (2019): 583.
>
> [Ref 3] Grouiller, Frederic, et al. "With or without spikes: localization of focal epileptic activity by simultaneous electroencephalography and functional magnetic resonance imaging." Brain 134.10 (2011): 2867-2886.
>
> [Ref 4] Heeger, David J., et al. "Spikes versus BOLD: what does neuroimaging tell us about neuronal activity?." Nature neuroscience 3.7 (2000): 631-633.
>
> [Ref 5] Smith, Arien J., et al. "Cerebral energetics and spiking frequency: the neurophysiological basis of fMRI." Proceedings of the National Academy of Sciences 99.16 (2002): 10765-10770.

---

> ### Author Response · Authors · 2025-11-21
> **Replies to other questions**
>
> Thank you for your high praise and valuable feedback on this paper. We have made corresponding revisions and additions to the original paper in response to your suggestions and shortcomings. The revised content in the original paper is marked in blue.
>
> **Q1:** How do you justify applying spike-based encoding to slow fMRI BOLD signals ( around 1 Hz)?
>
> **R1:** A response has already been received in the first round.
>
> **Q2:** How were τ_m, τ_s, and threshold values chosen?
>
> **R2:** The choices of τ_m, τ_s, and the threshold V_thre in our model have a certain neurophysiological basis. τ_m is the timescale of the neuronal membrane voltage response input, which is related to the type and physiological state of the neuron. τ_s controls the decay rate of the postsynaptic potential and is mainly used to simulate a rapidly decaying synaptic response. The threshold V_thre determines when the neuron fires a spike. The initial values of τ_m and τ_s reference the range of time constants of biological neurons and synapses [Ref 1~3]. To simplify the model and training process, V_thre is usually set to 1.
>
> [Ref 1] Gerstner, Wulfram, et al. Neuronal dynamics: From single neurons to networks and models of cognition. Cambridge University Press, 2014.
>
> [Ref 2] Dan, Yang, and Mu-Ming Poo. "Spike timing-dependent plasticity: from synapse to perception." Physiological reviews 86.3 (2006): 1033-1048.
>
> [Ref 3] Perez-Nieves, Nicolas, et al. "Neural heterogeneity promotes robust learning." Nature communications 12.1 (2021): 5791.
>
>
> **Q3:** How does Spike-DB differ fundamentally from earlier fMRI–SNN works (e.g., NeuCube, Kasabov et al., 2016)?
>
> **R3:** Thank you for your valuable feedback. Regarding the "first time" point, we have revised the relevant sections of the original paper (abstract, introduction, and conclusions). The fundamental difference between Spike-DB and NeuCube lies in our different approaches to modeling the temporal dynamics of the brain and their modeling objectives. The key differences are as follows:
>
> (1) Introduction of the spiking computation paradigm: Spike-DB encodes fMRI signals into spiking trains by introducing the spiking computation paradigm. This approach more accurately captures the temporal features of brain activity and the interactions between brain regions. In contrast, while NeuCube uses SNNs, it primarily focuses on learning spatiotemporal patterns and does not particularly emphasize spiking computation and temporal features.
>
> (2) Modeling of temporally driven relationships: Spike-DB learns the temporal driving relationships between anchored and target brain regions, revealing the dynamic dependencies and effective connections between brain regions. This allows Spike-DB to more accurately simulate causal relationships between brain regions. In contrast, NeuCube primarily learns spatiotemporal patterns through evolved SNNs, focusing on extracting spatiotemporal correlations from data rather than specifically modeling temporal dynamic characteristics.
>
> (3) Modeling goals and applications: Spike-DB not only focuses on learning spatiotemporal patterns, but also on capturing temporal driving relationships and causal dependencies between brain regions. This makes Spike-DB perform better in downstream tasks such as brain activity prediction, brain disease classification, and abnormal brain region identification. NeuCube, on the other hand, focuses more on the integration of spatiotemporal patterns and the learning of spatial information, emphasizing the identification of spatiotemporal structures between brain regions rather than in-depth modeling of causal relationships.

---

> ### Author Response · Authors · 2025-11-21
> **Replies to other questions**
>
> **Q4:** Will fix 89 anchor regions and predict only one be affected by fMRI’ high autocorrelation of near region?
>
> **R4:** In this paper, our goal is to achieve high-precision prediction of brain activity by learning the temporal driving relationships between some brain regions and other brain regions. Regarding your question about fixing 89 anchor regions and predicting only one target region, and whether this is affected by the high autocorrelation of nearby fMRI regions, we believe the model is flexible in this respect and does not rely entirely on the autocorrelation of a single target region. Although we selected one target region in the main experiment, the prediction of that target region does not rely entirely on autocorrelation, but also considers information from other regions. As shown in Figure 5 of the original paper, when $K$=80, i.e., when only 10 target brain regions remain (nearby regions are masked), Spike-DB still performs very well, with an average R² of 0.958 and an average RSE of 0.203. At this point, Spike-DB still outperforms BrainLM, BrainMass, and Brain-JEPA. When $K$=70, i.e., when only 20 target brain regions remain (nearby regions are masked), Spike-DB still maintains good performance, with an average R² of 0.949 and an average RSE of 0.214. At this point, Spike-DB still outperforms BrainLM and BrainMass. In summary, the Spike-DB model not only relies on the autocorrelation of a target region (even if only one target brain region remains), but also takes into account information from other regions.
>
> **Q5:** Please report standard deviations and significance tests to confirm the 1–2% improvements.
>
> **R5:** Thank you for your helpful comments. We have supplemented Tables 1 and 2 in the original text, including standard deviations and significance tests. The results show that our method is a significant improvement over the baseline method ($p<0.05$). The modifications to Tables 1 and 2 are as follows:
>
> **Table 1**. Results of fMRI time series prediction task achieved by all methods on six data types (i.e., FLE, TLE, NC-E, EMCI, SMC, and NC-A), averaged over five independent runs. ↑: the higher, the better; ↓: the lower, the better. The best results are in **bold**, * indicate significant improvements over previous methods ($p < 0.05$).
>
> | Method            | Metric    | FLE                   | TLE                   | NC-E                  | EMCI                  | SMC                   | NC-A                  |
> |-------------------|-----------|-----------------------|-----------------------|-----------------------|-----------------------|-----------------------|-----------------------|
> | **BrainLM**       | R² ↑      | 0.922 ± 0.043         | 0.948 ± 0.037         | 0.934 ± 0.048         | 0.907 ± 0.023         | 0.902 ± 0.043         | 0.936 ± 0.047         |
> |                   | RSE ↓     | 0.279 ± 0.045         | 0.266 ± 0.054         | 0.258 ± 0.035         | 0.305 ± 0.027         | 0.313 ± 0.048         | 0.254 ± 0.029         |
> | **BrainMass**     | R² ↑      | 0.938 ± 0.044         | 0.953 ± 0.054         | 0.958 ± 0.044         | 0.930 ± 0.038         | 0.929 ± 0.049         | 0.957 ± 0.026         |
> |                   | RSE ↓     | 0.249 ± 0.047         | 0.239 ± 0.029         | 0.204 ± 0.052         | 0.265 ± 0.022         | 0.267 ± 0.017         | 0.206 ± 0.042         |
> | **Brain-JEPA**    | R² ↑      | 0.952 ± 0.026         | 0.964 ± 0.026         | 0.967 ± 0.030         | 0.938 ± 0.039         | 0.943 ± 0.015         | 0.965 ± 0.023         |
> |                   | RSE ↓     | 0.219 ± 0.016         | 0.217 ± 0.026         | 0.182 ± 0.057         | 0.249 ± 0.035         | 0.240 ± 0.019         | 0.188 ± 0.037         |
> | **BrainSymphony** | R² ↑      | 0.963 ± 0.034         | 0.969 ± 0.040         | 0.974 ± 0.014         | 0.947 ± 0.034         | 0.948 ± 0.037         | 0.975 ± 0.026         |
> |                   | RSE ↓     | 0.192 ± 0.043         | 0.202 ± 0.042         | 0.162 ± 0.055         | 0.231 ± 0.029         | 0.228 ± 0.040         | 0.160 ± 0.047         |
> | **Spike-DB**      | R² ↑      | **0.972 ± 0.017***      | **0.979 ± 0.013***      | **0.983 ± 0.016***      | **0.954 ± 0.039***      | **0.957 ± 0.041**      | **0.981 ± 0.026***      |
> |                   | RSE ↓     | **0.168 ± 0.019**      | **0.184 ± 0.012***      | **0.131 ± 0.023***      | **0.213 ± 0.029***      | **0.207 ± 0.038***      | **0.137 ± 0.026***      |

---

> ### Author Response · Authors · 2025-11-21
> **Replies to other questions**
>
> **R5:** **Table 2**. Results of brain disease classification achieved by all methods on four binary classification tasks (i.e., FLE vs. NC-E, TLE vs. NC-E, EMCI vs. NC-A, and SMC vs. NC-A), averaged over five independent runs.
>
> | Method            | Metric    | FLE vs. NC-E          | TLE vs. NC-E          | EMCI vs. NC-A         | SMC vs. NC-A          |
> |-------------------|-----------|-----------------------|-----------------------|-----------------------|-----------------------|
> | **BrainLM**       | ACC(%)    | 84.26 ± 3.96          | 83.17 ± 4.19          | 78.79 ± 4.35          | 80.56 ± 3.65          |
> |                   | F1(%)     | 82.11 ± 2.61          | 77.92 ± 4.20          | 70.83 ± 3.20          | 75.29 ± 2.69          |
> | **STCAL**         | ACC(%)    | 86.11 ± 2.38          | 85.15 ± 4.27          | 81.82 ± 2.09          | 81.48 ± 3.01          |
> |                   | F1(%)     | 84.21 ± 4.13          | 81.93 ± 2.64          | 76.00 ± 4.48          | 75.61 ± 2.03          |
> | **BrainMass**     | ACC(%)    | 87.04 ± 2.79          | 86.14 ± 2.76          | 83.33 ± 3.44          | 81.69 ± 3.89          |
> |                   | F1(%)     | 87.76 ± 3.12          | 82.50 ± 2.94          | 78.43 ± 3.07          | 82.35 ± 3.04          |
> | **Brain-JEPA**    | ACC(%)    | 87.96 ± 2.72          | 87.13 ± 3.01          | 84.85 ± 2.71          | 83.10 ± 4.25          |
> |                   | F1(%)     | 86.59 ± 3.48          | 83.95 ± 4.04          | 81.48 ± 1.18          | 81.82 ± 3.28          |
> | **BrainSymphony** | ACC(%)    | 88.89 ± 2.24          | 88.12 ± 3.32          | 85.88 ± 3.45          | 84.51 ± 3.21          |
> |                   | F1(%)     | 87.76 ± 3.82          | 85.37 ± 3.55          | **86.21 ± 3.32**          | 83.58 ± 4.77          |
> | **Starformer**    | ACC(%)    | 89.81 ± 3.16          | 89.11 ± 3.54          | 86.36 ± 1.89          | 85.92 ± 2.63          |
> |                   | F1(%)     | 88.89 ± 3.06          | 86.75 ± 3.58          | 83.20 ± 3.88          | 85.71 ± 2.34          |
> | **Spike-DB**      | ACC(%)    | **91.67 ± 2.74***       | **90.10 ± 2.94***       | **87.88 ± 2.11***       | **86.95 ± 2.64***       |
> |                   | F1(%)     | **91.90 ± 2.37***       | **88.09 ± 2.70***       | 85.19 ± 3.01          | **87.32 ± 1.26***       |
>
> **Q6:** What classifier (MLP, linear, etc.) was used?
>
> **R6:** Thank you for your helpful comments. We have supplemented this in Appendix A.5 and drawn Figure 6 to further explain the classification method. Instead of fine-tuning the backbone network, we modified its structure according to the task shift (regression to classification), using the same encoder part and adding a classification head on top of it. For regression tasks, the loss function is shown in Equation 9 of the original paper. For binary classification tasks, we use the cross-entropy loss function, described as follows:
>
> Regarding the choice of classifier, since the main framework is in the spike space, traditional classifiers are no longer applicable. We will describe a vectorized implementation method for determining the predicted class based on the output spike sequence. Specifically, let $\vec{Z}[t]\in {{\mathbb{R}}^{m}}$ represent the spike fired by each output neuron over time, where $m$ is the number of output classes. Let $\vec{e}\in {{\mathbb{R}}^{m}}$ be the spike count for each output neuron, which can be obtained by summing $\vec{Z}[t]$ over $T$ time steps:
>
> $\vec{e}=\sum\limits_{t=0}^{T}{\vec{Z}[t]}$
>
> the index of $\vec{e}$ with the maximum count corresponds to the predicted class $\hat{y}$:
>
> $\hat{y}=\underset{i}{\mathop{\arg \max }}\,{{e}_{i}}$
>
> To facilitate understanding, we provide an example in Figure 6: Neuron 1 fires 6 spikes, Neuron 2 fires 3 spikes, and Neuron 3 fires 1 spike. Compared to Neurons 2 and 3, Neuron 1 fires the most spikes, corresponding to a predicted class of 1. For the binary classification task, we used the classification cross-entropy loss function ${{L}_{c}}$:
>
> ${{L}_{c}}=-(\hat{y}\log \hat{y}+(1-\hat{y})\log (1-y))$
>
> where $y$ is the true class.

---

> ### Author Response · Authors · 2025-11-21
> **Replies to other questions**
>
> **Q7:** Please evaluate Spike-DB on additional public datasets to demonstrate robustness and effectiveness.
>
> **R7:** Thank you for your constructive feedback. To further validate the robustness of Spike-DB, we have expanded the experiments in Appendix A.4 by adding a larger dataset from the Decoding Neurofeedback Project organized by the Japan Agency for Medical Research and Development (AMRD), which includes 1410 participants (due to time constraints, we extracted fMRI data from 849 participants). The results show that on large-scale datasets, Spike-DB not only performs exceptionally well in the fMRI time-series prediction task but also achieves leading performance in the brain disease classification task, demonstrating its superior ability and consistency in handling large-scale datasets. The expanded content in Appendix A.4 is as follows:
>
> To further verify the robustness of Spike-DB, we expand the experiment by incorporating a larger dataset. This dataset is sourced from the Decoding Neurofeedback Project organized by the Agency for Medical Research and Development (AMRD) in Japan, encompassing 3T MRI data (resting-state fMRI EPI images, corrupted T1-weighted images, and field maps) from 1410 subjects (620 patients and 790 normal controls). We extract resting-state fMRI data from this dataset, covering 230 time points, and uniformly categorize the subjects into disease-related and disease-free groups. Ultimately, the dataset includes 849 subjects, of whom 384 are multi-disease (MD) patients and 465 are normal controls (NC-M). Data preprocessing is consistent with the epilepsy and ADNI datasets.
>
> Table 4 reports the results of Spike-DB on the AMRD dataset for fMRI time series prediction. As shown in Table 4, Spike-DB maintains strong predictive power on large-scale datasets, with R² values of 0.964 and 0.973, and a significantly lower RSE value than other methods. Secondly, Table 5 reports the results of Spike-DB on the AMRD dataset for brain disease classification. Specifically, Spike-DB achieved an accuracy of 84.09\% and an F1 score of 85.12\%, both superior to other methods. This indicates that Spike-DB not only performs excellently in fMRI time series prediction but also achieves a leading position in brain disease classification, demonstrating its superior ability and consistency in handling large-scale datasets. This further validates the broad applicability and efficiency of Spike-DB in multiple brain disease tasks.
>
> **Table 4**. Results of fMRI time series prediction task achieved by all methods on the additional AMRD dataset (i.e., MD and NC-M data types), averaged over five independent runs.  ↑: the higher, the better; ↓: the lower, the better. The best results are in **bold**, * indicate significant improvements over previous methods ($p < 0.05$).
>
> | Method            | Metric                | MD                      | NC-M                    |
> |-------------------|-----------------------|-------------------------|-------------------------|
> | **BrainLM**       | R² ↑                  | 0.936 ± 0.041           | 0.931 ± 0.037           |
> |                   | RSE ↓                 | 0.253 ± 0.027           | 0.262 ± 0.034           |
> | **BrainMass**     | R² ↑                  | 0.945 ± 0.037           | 0.946 ± 0.036           |
> |                   | RSE ↓                 | 0.234 ± 0.049           | 0.232 ± 0.052           |
> | **Brain-JEPA**    | R² ↑                  | 0.951 ± 0.038           | 0.959 ± 0.026           |
> |                   | RSE ↓                 | 0.222 ± 0.012           | 0.203 ± 0.027           |
> | **BrainSymphony** | R² ↑                  | 0.957 ± 0.036           | 0.966 ± 0.017           |
> |                   | RSE ↓                 | 0.205 ± 0.035           | 0.183 ± 0.056           |
> | **Spike-DB**      | R² ↑                  | **0.964 ± 0.016***        | **0.973 ± 0.017***       |
> |                   | RSE ↓                 | **0.190 ± 0.030***        | **0.164 ± 0.010***       |
>
> **Table 5**. Results of brain disease classification achieved by all methods on the additional AMRD dataset (i.e., MD vs. NC-M), averaged over five independent runs.
>
> | Method            | ACC(%)                | F1(%)                   |
> |-------------------|-----------------------|-------------------------|
> | **BrainLM**       | 74.69 ± 3.39          | 73.89 ± 2.84            |
> | **STCAL**         | 77.78 ± 3.40          | 72.73 ± 1.97            |
> | **BrainMass**     | 80.86 ± 4.94          | 74.14 ± 2.99            |
> | **Brain-JEPA**    | 82.08 ± 2.62          | 77.37 ± 2.06            |
> | **BrainSymphony** | 83.33 ± 1.41          | 78.05 ± 4.17            |
> | **Starformer**    | 83.82 ± 2.59          | 81.08 ± 3.09            |
> | **Spike-DB**      | **84.09 ± 2.64***       | **85.12 ± 2.89***        |

---

> ### Author Response · Authors · 2025-11-21
> **Replies to other questions**
>
> **Q8:** How does your perturbation-based EC relate to established causal models (e.g., DCM, Granger)?
>
> **R8:** The efficient connectivity perturbation strategy proposed in this paper is based on the Neural Perturbational Inference (NPI) method proposed by Luo et al. [Ref 1]. It is worth noting that our Spike-DB model operates entirely in spike space; therefore, unlike NPI, we incorporate a Gaussian pulse-based perturbation strategy to enable efficient connectivity inference in spike space. However, the fundamental perturbation strategy remains the same as that of NPI. To our knowledge, Luo et al. have compared the performance of NPI with Granger and DCM on multiple datasets. We are currently encountering some difficulties in DCM validation, mainly because DCM usually requires task-state data, while our current research focuses on resting-state data. Therefore, we plan to continue exploring and validating DCM in future work.
>
> [Ref 1] Luo, Zixiang, et al. "Mapping effective connectivity by virtually perturbing a surrogate brain." Nature Methods (2025): 1-10.
>
> **Q9:** Will you release code, model weights, and preprocessing scripts to enable replication?
>
> **R9:** Since the current score for this paper is not ideal, we plan to release the relevant code and pre-trained model after the paper is accepted.

---

> > ### Comment · Reviewer_dTth · 2025-11-26
> > **I have raised my score to 6**
> >
> > Thank you for the clarification and the revision. I have increased my score to 6.

---

> > > ### Author Response · Authors · 2025-11-26
> > >
> > > Thank you so much for acknowledging our response and for your decision to raise the rating! We sincerely appreciate your constructive feedback during the review process, which played a vital role in improving the quality of our work. Thank you again for your involvement and support.

---

### Official Review · Reviewer_94Co · 2025-10-26

**Soundness:** 2
**Presentation:** 2
**Contribution:** 1
**Rating:** 2
**Confidence:** 4

**Summary:**

This paper introduces Spike-DB, a framework that converts fMRI time series into discrete “spike” events and trains spike Transformer with a masked-region prediction objective. The learned embeddings are then used for downstream tasks such as detecting abnormal brain regions and classifying neurological diseases. Experiments on two disease datasets show improvements over selected baselines, and ablations aim to assess the contribution of the spike encoding were performed.

**Strengths:**

Framing fMRI analysis as digital spiking signal is a fresh and interesting direction.

**Weaknesses:**

* Because fMRI measures BOLD signals rather than direct neural spiking, the biological interpretability of the discrete spikes remains unclear. It would be valuable to connect the event representation to known hemodynamic dynamics or evaluate on a modality closer to neural spiking (e.g., iEEG).

* The need to train separate embedding models for frontal lobe epilepsy, temporal lobe epilepsy, different levels of cognitive impairment, and healthy controls for each dataset suggests limited cross-condition generalization. This raises concerns about whether the method captures broadly generalizable biological features versus dataset- or subtype-specific cues.

**Questions:**

* Is the train/test spilt subject-based or data-based? What was the training data used the downstream classifiers? What is the training procedure for downstream tasks? If the same training data was used in the pretrain and decoding, how does an end-to-end model trained directly for the downstream task (without the spike pretext stage) perform relative to Spike-DB?

* How exactly was the “no spiking” ablation implemented? Did you remove only the spike encoding while keeping the Spike Transformer?

---

> ### Author Response · Authors · 2025-11-15
> **Response regarding the biological rationale of this Paper**
>
> We thank the four reviewers for their insightful comments on this paper. We note that all four reviewers pointed out the infeasibility of encoding fMRI signals into spike trains. Therefore, we are addressing this issue first. Regarding other related experiments and questions, we will respond as soon as possible before the deadline and will do our best to meet the reviewers' expectations.
>
> First, our method does not directly convert fMRI signals into high-frequency spike trains at the neuron level; instead, it converts brain region-level fMRI signals into spike representations. These spike representations do not capture neuronal-level firing behavior, but rather efficiently encode brain region activity through spike trains. To avoid misunderstanding, we have revised the relevant content in the original paper.
>
> Our motivation for converting fMRI signals into spike representations stems from the classic hemodynamic response function (HRF), a mathematical model describing the dynamic response of cerebral blood flow to neural activity. Specifically, the core of the HRF model lies in its description of the impulse response of blood flow after stimulation; that is, after activation of a brain region, the blood flow response does not simply increase linearly, but exhibits a fluctuating pattern, initially rising and then gradually declining. This response pattern is similar to the spike firing process of neurons, where neurons rapidly fire in the form of spikes upon receiving stimulation. Although the timescale of the blood flow response is relatively long, it still captures the spike-like influence of neural activity on blood flow. In this way, we can draw an analogy between changes in fMRI signals and the spike firing of neurons, and use SNNs to simulate this temporal response. According to [Ref 1], the BOLD spike response $y(i,t)$ of fMRI signals can be represented as the convolution of the activity-induced signal and the HRF:
>
> $y(i,t)=\sum{c(i)\delta (t)\cdot h(t)+\sum{\beta (i)n(t)+\mu (i,t)}}$
>
> where $t$ is time, $c(i)$ is the amplitude of neural activity on the  $i$-th voxel, $h(t)$ is the HRF function, $\delta (t)$ is the Dirac-delta function, $n(t)$ is the known perturbation regression (movement or low-frequency drift, etc.), $\beta (i)$ is the correlation weight, and $\mu (i,t)$ is independently distributed Gaussian noise.
>
> Current research has converted fMRI signals into spike representations for further analysis. For example, Deco et al. [Ref 2] proposed the Brain Songs framework to explore the relevant time scales of human brain signals, demonstrating the effectiveness of encoding fMRI signals into spike trains in capturing brain functional connectivity. Previous studies [Ref 3-5] have shown a linear relationship between the BOLD signal of fMRI and the neuronal firing rate. Compared with these studies, our innovation lies in using SNNs to generate spike representations more naturally, rather than first inferring the BOLD signal as a continuous variable through a model and then converting it into pseudo-spike events, as is done in traditional methods.
>
> It is important to note that capturing neuronal-level firing behavior is indeed a very complex and challenging task, which cannot be accomplished solely with non-invasive devices such as fMRI and EEG. We anticipate that with advancements in data acquisition technologies, our models will be able to truly characterize the fine-grained activity at the neuronal level in the brain.
>
> [Ref 1] Karahanoğlu F I, Caballero-Gaudes C, Lazeyras F, et al. Total activation: fMRI deconvolution through spatio-temporal regularization[J]. Neuroimage, 2013, 73: 121-134.
>
> [Ref 2] Deco, Gustavo, Josephine Cruzat, and Morten L. Kringelbach. "Brain songs framework used for discovering the relevant timescale of the human brain." Nature communications 10.1 (2019): 583.
>
> [Ref 3] Grouiller, Frederic, et al. "With or without spikes: localization of focal epileptic activity by simultaneous electroencephalography and functional magnetic resonance imaging." Brain 134.10 (2011): 2867-2886.
>
> [Ref 4] Heeger, David J., et al. "Spikes versus BOLD: what does neuroimaging tell us about neuronal activity?." Nature neuroscience 3.7 (2000): 631-633.
>
> [Ref 5] Smith, Arien J., et al. "Cerebral energetics and spiking frequency: the neurophysiological basis of fMRI." Proceedings of the National Academy of Sciences 99.16 (2002): 10765-10770.

---

> ### Author Response · Authors · 2025-11-21
> **Replies to other questions**
>
> Thank you for your high praise and valuable feedback on this paper. We have made corresponding revisions and additions to the original paper in response to your suggestions and shortcomings. The revised content in the original paper is marked in blue.
>
> **Q1:** The need to train separate embedding models for frontal lobe epilepsy, temporal lobe epilepsy, different levels of cognitive impairment, and healthy controls for each dataset suggests limited cross-condition generalization.
>
> **R1:** In this paper, for the fMRI prediction task, we included two datasets: epilepsy (FLE, TLE, NC-E) and ADNI (EMCI, SMC, NC-A), which we divided into six data types (FLE, TLE, EMCI, SMC, NC-E, and NC-A). We trained a separate model for each dataset to better capture the unique characteristics of each brain disease. Different diseases or subtypes have their own unique pathological mechanisms and neurophysiological features. By training the model separately, we can optimize the performance of each dataset to the greatest extent, thereby improving prediction accuracy. Furthermore, for the brain disease classification task, in the current research context, similar to most studies, our method still focuses on binary classification problems. This is because many brain diseases, especially epilepsy and other subtypes, often exhibit significant individual differences and complex pathological mechanisms, making multi-class classification tasks more challenging. With increasing data volume and continuous optimization of research methods, we hope to gradually extend our method to more complex multi-class classification tasks in the future.
>
> **Q2:** Is the train/test spilt subject-based or data-based?
>
> **R2:** In all tasks, the division of the training and test sets is based on the subjects rather than the data. The training data used in the downstream task mainly includes four groups: FLE vs. NC-E, TLE vs. NC-E, EMCI vs. NC-A, and SMC vs. NC-A. To transition from a regression task to a classification task, we modified the model architecture, using the same encoder part and adding a classification head on top of it. Specifically, fMRI data is first encoded into spike trains using spiking neuron layer, then pre-trained using an encoder (anchor, target, and predictor), and used as a feature extractor for the downstream task. Finally, these extracted spike features are input into the classifier for brain disease classification. We have revised Section 3.1 and added details regarding the downstream brain disease classification task in Appendix A.5.
>
> Regarding the choice of classifier, since the main framework is in the spike space, traditional classifiers are no longer applicable. We will describe a vectorized implementation method for determining the predicted class based on the output spike sequence. Specifically, let $\vec{Z}[t]\in {{\mathbb{R}}^{m}}$ represent the spike fired by each output neuron over time, where $m$ is the number of output classes. Let $\vec{e}\in {{\mathbb{R}}^{m}}$ be the spike count for each output neuron, which can be obtained by summing $\vec{Z}[t]$ over $T$ time steps:
>
> $\vec{e}=\sum\limits_{t=0}^{T}{\vec{Z}[t]}$
>
> the index of $\vec{e}$ with the maximum count corresponds to the predicted class $\hat{y}$:
>
> $\hat{y}=\underset{i}{\mathop{\arg \max }}\,{{e}_{i}}$
>
> To facilitate understanding, we provide an example in Figure 6: Neuron 1 fires 6 spikes, Neuron 2 fires 3 spikes, and Neuron 3 fires 1 spike. Compared to Neurons 2 and 3, Neuron 1 fires the most spikes, corresponding to a predicted class of 1. For the binary classification task, we used the classification cross-entropy loss function ${{L}_{c}}$:
>
> ${{L}_{c}}=-(\hat{y}\log \hat{y}+(1-\hat{y})\log (1-y))$
>
> where $y$ is the true class.
>
> **Q3:** How exactly was the “no spiking” ablation implemented? Did you remove only the spike encoding while keeping the Spike Transformer?
>
> **R3:** In the ablation study without spikes, we removed the spiking neuron layers from all components, thus eliminating not only the spiking coding mechanism but also degenerating the Spike Transformer into a regular Transformer. We have supplemented Section 3.6 of the original paper.

---

### Official Review · Reviewer_us19 · 2025-10-30

**Soundness:** 3
**Presentation:** 3
**Contribution:** 3
**Rating:** 6
**Confidence:** 3

**Summary:**

This paper presents a new fundamental model, the Spike-based Digital Brain (Spike-DB), which addresses the limitations of continuous-valued frameworks in brain modeling. A key contribution of this paper is the first-time application of the spike computing paradigm to fMRI time series analysis. The model converts blood oxygenation level-dependent signals into spike trains using an IIR filter-based SNN and is trained via a self-supervised, predictive mechanism (learning from 'anchor' to 'target' regions) on a Spike Transformer architecture.

The authors validate Spike-DB using epilepsy and ADNI datasets, demonstrating state-of-the-art (SOTA) performance in both fMRI time series prediction and brain disease classification when compared against several leading fundamental and specialized models. The model's clinical utility is further explored through downstream tasks, where it successfully identified abnormal brain regions and inferred effective connectivity patterns that were consistent with existing medical research.

**Strengths:**

- The paper is well-written, making it easy to follow and understand.
- The authors define clear research questions (RQs) and validate claims thoroughly via SOTA comparison (RQ1), clinical interpretation (RQ2), and ablation studies (RQ3).
- The paper's novel contribution, being the first to apply spike computing to fMRI modeling, is not merely a conceptual claim. The authors successfully validate this approach by demonstrating that their proposed paradigm, when practically embedded in the Spike-DB model, leads directly to new SOTA performance over existing methods.
- The model shows meaningful clinical utility by identifying abnormal brain regions and connectivity patterns that are well-supported by existing medical research.

**Weaknesses:**

- The idea of converting slow, hemodynamic fMRI signals into discrete, fast neural spikes may lack sufficient biological justification and could be conceptually problematic.
- The model's claimed flexibility (implied in Fig. 1) is contradicted by its reliance on an extreme parameter setting. Optimal performance is achieved when $K$ = 89 out of 90 total ROIs, leaving 'only one target region'. This suggests the model is highly optimized for a very specific 'predict-one-from-all' task, not a general masking strategy. This high sensitivity is confirmed by the ablation study, which shows performance 'deteriorates significantly' as K decreases ($K$ < 45, as shown in subsection 3.6), raising concerns about the model's practical flexibility.
- To fully understand the practical trade-offs of the proposed method, an analysis of the computational overhead is required. The Spike Transformer and SNN layers are typically resource-intensive when simulated on standard GPUs (like the RTX 4090 used) compared to the standard Transformer baselines. A discussion or comparison of training time and resource usage was not included. This analysis would be valuable in future work, as it would provide a more complete picture of the costs associated with the SOTA performance reported in Tables 1 & 2.

**Questions:**

- Q1. How can we be certain that the model's superior performance truly stems from being 'closer to the biological nervous system', and not simply from the IIR filter-based SNN acting as a highly effective non-linear feature extractor for the specific temporal dynamics of BOLD data? I'm curious about the authors' opinions.
- Q2. To better assess the practical trade-offs, could the authors provide a comparison of computational costs (e.g., training time, resource usage) against the baselines? This information would be a valuable addition for evaluating the model's real-world applicability.

**Details Of Ethics Concerns:**

None.

---

> ### Author Response · Authors · 2025-11-15
> **Response regarding the biological rationale of this Paper**
>
> We thank the four reviewers for their insightful comments on this paper. We note that all four reviewers pointed out the infeasibility of encoding fMRI signals into spike trains. Therefore, we are addressing this issue first. Regarding other related experiments and questions, we will respond as soon as possible before the deadline and will do our best to meet the reviewers' expectations.
>
> First, our method does not directly convert fMRI signals into high-frequency spike trains at the neuron level; instead, it converts brain region-level fMRI signals into spike representations. These spike representations do not capture neuronal-level firing behavior, but rather efficiently encode brain region activity through spike trains. To avoid misunderstanding, we have revised the relevant content in the original paper.
>
> Our motivation for converting fMRI signals into spike representations stems from the classic hemodynamic response function (HRF), a mathematical model describing the dynamic response of cerebral blood flow to neural activity. Specifically, the core of the HRF model lies in its description of the impulse response of blood flow after stimulation; that is, after activation of a brain region, the blood flow response does not simply increase linearly, but exhibits a fluctuating pattern, initially rising and then gradually declining. This response pattern is similar to the spike firing process of neurons, where neurons rapidly fire in the form of spikes upon receiving stimulation. Although the timescale of the blood flow response is relatively long, it still captures the spike-like influence of neural activity on blood flow. In this way, we can draw an analogy between changes in fMRI signals and the spike firing of neurons, and use SNNs to simulate this temporal response. According to [Ref 1], the BOLD spike response $y(i,t)$ of fMRI signals can be represented as the convolution of the activity-induced signal and the HRF:
>
> $y(i,t)=\sum{c(i)\delta (t)\cdot h(t)+\sum{\beta (i)n(t)+\mu (i,t)}}$
>
> where $t$ is time, $c(i)$ is the amplitude of neural activity on the  $i$-th voxel, $h(t)$ is the HRF function, $\delta (t)$ is the Dirac-delta function, $n(t)$ is the known perturbation regression (movement or low-frequency drift, etc.), $\beta (i)$ is the correlation weight, and $\mu (i,t)$ is independently distributed Gaussian noise.
>
> Current research has converted fMRI signals into spike representations for further analysis. For example, Deco et al. [Ref 2] proposed the Brain Songs framework to explore the relevant time scales of human brain signals, demonstrating the effectiveness of encoding fMRI signals into spike trains in capturing brain functional connectivity. Previous studies [Ref 3-5] have shown a linear relationship between the BOLD signal of fMRI and the neuronal firing rate. Compared with these studies, our innovation lies in using SNNs to generate spike representations more naturally, rather than first inferring the BOLD signal as a continuous variable through a model and then converting it into pseudo-spike events, as is done in traditional methods.
>
> It is important to note that capturing neuronal-level firing behavior is indeed a very complex and challenging task, which cannot be accomplished solely with non-invasive devices such as fMRI and EEG. We anticipate that with advancements in data acquisition technologies, our models will be able to truly characterize the fine-grained activity at the neuronal level in the brain.
>
> [Ref 1] Karahanoğlu F I, Caballero-Gaudes C, Lazeyras F, et al. Total activation: fMRI deconvolution through spatio-temporal regularization[J]. Neuroimage, 2013, 73: 121-134.
>
> [Ref 2] Deco, Gustavo, Josephine Cruzat, and Morten L. Kringelbach. "Brain songs framework used for discovering the relevant timescale of the human brain." Nature communications 10.1 (2019): 583.
>
> [Ref 3] Grouiller, Frederic, et al. "With or without spikes: localization of focal epileptic activity by simultaneous electroencephalography and functional magnetic resonance imaging." Brain 134.10 (2011): 2867-2886.
>
> [Ref 4] Heeger, David J., et al. "Spikes versus BOLD: what does neuroimaging tell us about neuronal activity?." Nature neuroscience 3.7 (2000): 631-633.
>
> [Ref 5] Smith, Arien J., et al. "Cerebral energetics and spiking frequency: the neurophysiological basis of fMRI." Proceedings of the National Academy of Sciences 99.16 (2002): 10765-10770.

---

> ### Author Response · Authors · 2025-11-21
> **Replies to other questions**
>
> Thank you for your high praise and valuable feedback on this paper. We have made corresponding revisions and additions to the original paper in response to your suggestions and shortcomings. The revised content in the original paper is marked in blue.
>
> **Q1:** The model's claimed flexibility is contradicted by its reliance on an extreme parameter setting.
>
> **R1:** In this paper, our goal is to achieve high-precision prediction of brain activity by learning the temporal driving relationships between some brain regions and other brain regions. To avoid misunderstanding, we have supplemented Section 3.6 of the original text. In this paper, based on ablation studies (see Figure 5 in the original paper), the Spike-DB model achieves optimal performance when $K$=89, i.e., only one target brain region remains. However, when $K$=70, i.e., 20 target brain regions remain, the Spike-DB model still performs well. For example, when $K$=70, the average R² is 0.949, and the average RSE is 0.214. A comparison with Table 1 shows that Spike-DB still outperforms BrainLM and BrainMass. Furthermore, in practical applications, the number of predicted target brain regions is usually kept below 30. When K<45, i.e., accurately predicting multiple target brain regions from less than half of the brain region data, remains a challenge for current technologies. We also look forward to achieving this goal as soon as possible through continuous optimization of the model.
>
> **Q2:** It is closer to the biological nervous system.
>
> **R2:** A response has already been given in the first round.
>
> **Q3:** The computational cost of the model.
>
> **R3:** Thank you for your valuable feedback. Based on your comments, we have supplemented the computational cost of the model in Appendix A.6, including the number of parameters and training time for all methods. Specifically, as shown in Table 6, the Spike-DB method demonstrates significant advantages in terms of parameter count and training time by introducing a spiking mechanism. Compared to other methods, Spike-DB has only 2.7 M parameters, far lower than methods such as Brain-JEPA (22.0 M) and BrainMass (14.4 M), while the training time per epoch is only 96.73 seconds, the shortest of all methods. This is thanks to the introduction of the spiking mechanism, which effectively reduces computational complexity and memory consumption by simulating the spiking firing pattern of neurons, enabling Spike-DB to significantly improve training efficiency while maintaining performance. The expanded content in Appendix A.6 is as follows:
>
> As shown in Table 6, the Spike-DB method demonstrates significant advantages in terms of parameter count and training time by introducing a spiking mechanism. Compared to other methods, Spike-DB has only 2.7 M parameters, far lower than Brain-JEPA (22.0 M) and BrainMass (14.4 M), while the training time per epoch is only 96.73 s, the shortest of all methods. This is thanks to the introduction of the spiking mechanism, which effectively reduces computational complexity and memory consumption by simulating the spiking firing pattern of neurons, allowing Spike-DB to significantly improve training efficiency while maintaining performance.
>
> **Table 6**. The total number of parameters for all methods and their average training time per epoch (based on the average of six data types: FLE, TLE, NC-E, EMCI, SMC, and NC-A).
>
> | Method            | Parameters (M)   | Training time (s)  |
> |-------------------|------------------|--------------------|
> | **BrainLM**       | 13.0             | 163.59             |
> | **BrainMass**     | 14.4             | 284.65             |
> | **Brain-JEPA**    | 22.0             | 127.81             |
> | **BrainSymphony** | 2.0              | 114.18             |
> | **Spike-DB**      | 2.7              | 96.73              |

---

> > ### Comment · Reviewer_us19 · 2025-11-26
> >
> > Thank you for the clear rebuttal. However, I noticed a minor error in your response "when $K$=70, i.e., 30 target brain regions remain." Given the total of 90 ROIs, $K$=70 should correspond to 20 target regions, not 30. While this appears to be a small mistake, careful attention to detail is necessary to properly finalize the paper. Overall, all my concerns have been addressed. Please ensure that the modifications currently marked in blue are reflected in the final version. I also look forward to seeing the challenges you acknowledged developed into advanced research in your future work.

---

> > > ### Author Response · Authors · 2025-11-26
> > >
> > > Thank you very much for your support and valuable feedback! We have corrected the errors and updated the relevant sections.
> > >
> > > Although the average score for this paper is not ideal at present, regardless of the final acceptance result, we sincerely appreciate your recognition and constructive comments in the initial review, which is a great encouragement to us.

---

### Official Review · Reviewer_UykD · 2025-10-31

**Soundness:** 3
**Presentation:** 3
**Contribution:** 3
**Rating:** 4
**Confidence:** 3

**Summary:**

This paper proposes Spike-Based Digital Brain (Spike-DB), a novel computational framework designed to model brain activity by integrating spike-based neural computation with fMRI time-series prediction. The method encodes continuous fMRI signals into discrete spike trains using an IIR-based spiking neuron model, aiming to capture temporal dependencies more effectively and in a biologically inspired manner. Spike-DB learns to predict the activity of target brain regions from selected anchor regions, enabling a self-supervised learning paradigm that mimics information flow in the brain. The trained model is further applied to several downstream tasks, including brain disease classification, abnormal brain region identification, and effective connectivity inference.

**Strengths:**

This paper presents an interesting idea of using spike-based computation for modeling brain activity, which offers a fresh perspective compared to standard fMRI analysis methods. The proposed framework, Spike-DB, is technically sound and reasonably well explained, with clear figures and equations that make the approach understandable. The authors also provide experimental results showing small but consistent improvements over existing baselines, suggesting that the method has potential.

**Weaknesses:**

The main weaknesses of this paper stem from issues of biological plausibility, reproducibility, and evaluation scope. Most notably, the conversion of low-temporal-resolution fMRI BOLD signals into high-frequency spike trains is conceptually problematic, as fMRI data reflect slow, indirect hemodynamic responses rather than true neural spiking activity. This makes the biological interpretation of the model questionable and undermines its claim of being a biologically inspired “digital brain.” Additionally, the study lacks reproducibility, since no implementation code, pretrained models, or detailed preprocessing procedures are provided, making it difficult for others to verify the reported results. Finally, the experiments are confined to two relatively small datasets (epilepsy and ADNI), limiting the model’s generalizability and raising concerns about potential overfitting and the robustness of the findings.

**Questions:**

1. Biological Implausibility of Spike Conversion:
The transformation of low-temporal-resolution fMRI BOLD signals into high-frequency spike trains is not physiologically realistic. Since fMRI measures slow hemodynamic responses rather than actual neural spiking activity, this conversion undermines the biological credibility of the proposed “spike-based” framework and weakens the paper’s neuroscientific claims.

2. Limited Reproducibility:
The paper lacks publicly available code, pretrained models, and detailed data preprocessing information. Without these resources, the reported results cannot be independently verified, which significantly limits the work’s transparency and scientific reproducibility.

3. Restricted Evaluation Scope:
The experiments are confined to two relatively small datasets (epilepsy and ADNI), without cross-dataset or large-scale validation. This narrow evaluation raises concerns about the model’s robustness, risk of overfitting, and generalization to broader neuroimaging domains.

---

> ### Author Response · Authors · 2025-11-15
> **Response regarding the biological rationale of this Paper**
>
> We thank the four reviewers for their insightful comments on this paper. We note that all four reviewers pointed out the infeasibility of encoding fMRI signals into spike trains. Therefore, we are addressing this issue first. Regarding other related experiments and questions, we will respond as soon as possible before the deadline and will do our best to meet the reviewers' expectations.
>
> First, our method does not directly convert fMRI signals into high-frequency spike trains at the neuron level; instead, it converts brain region-level fMRI signals into spike representations. These spike representations do not capture neuronal-level firing behavior, but rather efficiently encode brain region activity through spike trains. To avoid misunderstanding, we have revised the relevant content in the original paper.
>
> Our motivation for converting fMRI signals into spike representations stems from the classic hemodynamic response function (HRF), a mathematical model describing the dynamic response of cerebral blood flow to neural activity. Specifically, the core of the HRF model lies in its description of the impulse response of blood flow after stimulation; that is, after activation of a brain region, the blood flow response does not simply increase linearly, but exhibits a fluctuating pattern, initially rising and then gradually declining. This response pattern is similar to the spike firing process of neurons, where neurons rapidly fire in the form of spikes upon receiving stimulation. Although the timescale of the blood flow response is relatively long, it still captures the spike-like influence of neural activity on blood flow. In this way, we can draw an analogy between changes in fMRI signals and the spike firing of neurons, and use SNNs to simulate this temporal response. According to [Ref 1], the BOLD spike response $y(i,t)$ of fMRI signals can be represented as the convolution of the activity-induced signal and the HRF:
>
> $y(i,t)=\sum{c(i)\delta (t)\cdot h(t)+\sum{\beta (i)n(t)+\mu (i,t)}}$
>
> where $t$ is time, $c(i)$ is the amplitude of neural activity on the  $i$-th voxel, $h(t)$ is the HRF function, $\delta (t)$ is the Dirac-delta function, $n(t)$ is the known perturbation regression (movement or low-frequency drift, etc.), $\beta (i)$ is the correlation weight, and $\mu (i,t)$ is independently distributed Gaussian noise.
>
> Current research has converted fMRI signals into spike representations for further analysis. For example, Deco et al. [Ref 2] proposed the Brain Songs framework to explore the relevant time scales of human brain signals, demonstrating the effectiveness of encoding fMRI signals into spike trains in capturing brain functional connectivity. Previous studies [Ref 3-5] have shown a linear relationship between the BOLD signal of fMRI and the neuronal firing rate. Compared with these studies, our innovation lies in using SNNs to generate spike representations more naturally, rather than first inferring the BOLD signal as a continuous variable through a model and then converting it into pseudo-spike events, as is done in traditional methods.
>
> It is important to note that capturing neuronal-level firing behavior is indeed a very complex and challenging task, which cannot be accomplished solely with non-invasive devices such as fMRI and EEG. We anticipate that with advancements in data acquisition technologies, our models will be able to truly characterize the fine-grained activity at the neuronal level in the brain.
>
> [Ref 1] Karahanoğlu F I, Caballero-Gaudes C, Lazeyras F, et al. Total activation: fMRI deconvolution through spatio-temporal regularization[J]. Neuroimage, 2013, 73: 121-134.
>
> [Ref 2] Deco, Gustavo, Josephine Cruzat, and Morten L. Kringelbach. "Brain songs framework used for discovering the relevant timescale of the human brain." Nature communications 10.1 (2019): 583.
>
> [Ref 3] Grouiller, Frederic, et al. "With or without spikes: localization of focal epileptic activity by simultaneous electroencephalography and functional magnetic resonance imaging." Brain 134.10 (2011): 2867-2886.
>
> [Ref 4] Heeger, David J., et al. "Spikes versus BOLD: what does neuroimaging tell us about neuronal activity?." Nature neuroscience 3.7 (2000): 631-633.
>
> [Ref 5] Smith, Arien J., et al. "Cerebral energetics and spiking frequency: the neurophysiological basis of fMRI." Proceedings of the National Academy of Sciences 99.16 (2002): 10765-10770.

---

> ### Author Response · Authors · 2025-11-21
> **Replies to other questions**
>
> Thank you for your high praise and valuable feedback on this paper. We have made corresponding revisions and additions to the original paper in response to your suggestions and shortcomings. The revised content in the original paper is marked in blue.
>
> **Q1**: Biological Implausibility of Spike Conversion？
>
> **R1**: A response has already been given in the first round.
>
>
> **Q2**: Limited Reproducibility?
>
> **R2**: Since the current score for this paper is not ideal, we plan to release the relevant code and pre-trained model after the paper is accepted. Meanwhile, detailed information regarding fMRI data preprocessing has been provided in Appendix A.1. The modified contents in Appendix A.1 are as follows:
>
> Data are collected using a Siemens Trio 3 T MRI scanner. fMRI scans are acquired using a single-shot, gradient-echo planar imaging sequence. Scan parameters are as follows: repetition time = 2000 $ms$, echo time = 30 $ms$, flip angle = 90°, 30 transverse slices, field of view (FOV) = 240 $\times $ 240 $m{{m}^{2}}$, slice thickness = 4 $mm$, interstice gap = 0.4 $mm$, voxel size = 3.75 $\times $ 3.75 $\times $ 3.75 $m{{m}^{3}}$.
>
> We use SPM12 version 2.0 of the DPARSF toolkit to preprocess all fMRI data in the ADNI and epilepsy datasets. First, the initial functional time series are slice-time acquisition corrected, re-aligned, and normalized to the echo-planar imaging (EPI) template. Subsequently, detrending is performed to eliminate sources of spurious variance (i.e., six head motion parameters, averaged signals from cerebrospinal fluid and white matter, and global brain signal), followed by bandpass filtering (0.01 to 0.08 Hz) on the time series data. Finally, the data are divided into 90 regions of interest (ROIs) based on the automatic anatomical Labeling (AAL) template. The functional data for each subject in the epilepsy dataset contain 240 time points, and the functional data for each subject in the ADNI dataset contain 197 time points.

---

> ### Author Response · Authors · 2025-11-21
> **Replies to other questions**
>
> **Q3**: Restricted Evaluation Scope?
>
> **R3**: Thank you for your constructive feedback. To further validate the robustness of Spike-DB, we have expanded the experiments in Appendix A.4 by adding a larger dataset from the Decoding Neurofeedback Project organized by the Japan Agency for Medical Research and Development (AMRD), which includes 1410 participants (due to time constraints, we extracted fMRI data from 849 participants). The results show that on large-scale datasets, Spike-DB not only performs exceptionally well in the fMRI time-series prediction task but also achieves leading performance in the brain disease classification task, demonstrating its superior ability and consistency in handling large-scale datasets. The expanded content in Appendix A.4 is as follows:
>
> To further verify the robustness of Spike-DB, we expand the experiment by incorporating a larger dataset. This dataset is sourced from the Decoding Neurofeedback Project organized by the Agency for Medical Research and Development (AMRD) in Japan, encompassing 3T MRI data (resting-state fMRI EPI images, corrupted T1-weighted images, and field maps) from 1410 subjects (620 patients and 790 normal controls). We extract resting-state fMRI data from this dataset, covering 230 time points, and uniformly categorize the subjects into disease-related and disease-free groups. Ultimately, the dataset includes 849 subjects, of whom 384 are multi-disease (MD) patients and 465 are normal controls (NC-M). Data preprocessing is consistent with the epilepsy and ADNI datasets.
>
> Table 4 reports the results of Spike-DB on the AMRD dataset for fMRI time series prediction. As shown in Table 4, Spike-DB maintains strong predictive power on large-scale datasets, with R² values of 0.964 and 0.973, and a significantly lower RSE value than other methods. Secondly, Table 5 reports the results of Spike-DB on the AMRD dataset for brain disease classification. Specifically, Spike-DB achieved an accuracy of 84.09\% and an F1 score of 85.12\%, both superior to other methods. This indicates that Spike-DB not only performs excellently in fMRI time series prediction but also achieves a leading position in brain disease classification, demonstrating its superior ability and consistency in handling large-scale datasets. This further validates the broad applicability and efficiency of Spike-DB in multiple brain disease tasks.
>
> **Table 4**. Results of fMRI time series prediction task achieved by all methods on the additional AMRD dataset (i.e., MD and NC-M data types), averaged over five independent runs.  ↑: the higher, the better; ↓: the lower, the better. The best results are in **bold**, * indicate significant improvements over previous methods ($p < 0.05$).
>
> | Method            | Metric                | MD                      | NC-M                    |
> |-------------------|-----------------------|-------------------------|-------------------------|
> | **BrainLM**       | R² ↑                  | 0.936 ± 0.041           | 0.931 ± 0.037           |
> |                   | RSE ↓                 | 0.253 ± 0.027           | 0.262 ± 0.034           |
> | **BrainMass**     | R² ↑                  | 0.945 ± 0.037           | 0.946 ± 0.036           |
> |                   | RSE ↓                 | 0.234 ± 0.049           | 0.232 ± 0.052           |
> | **Brain-JEPA**    | R² ↑                  | 0.951 ± 0.038           | 0.959 ± 0.026           |
> |                   | RSE ↓                 | 0.222 ± 0.012           | 0.203 ± 0.027           |
> | **BrainSymphony** | R² ↑                  | 0.957 ± 0.036           | 0.966 ± 0.017           |
> |                   | RSE ↓                 | 0.205 ± 0.035           | 0.183 ± 0.056           |
> | **Spike-DB**      | R² ↑                  | **0.964 ± 0.016***        | **0.973 ± 0.017***       |
> |                   | RSE ↓                 | **0.190 ± 0.030***        | **0.164 ± 0.010***       |
>
> **Table 5**. Results of brain disease classification achieved by all methods on the additional AMRD dataset (i.e., MD vs. NC-M), averaged over five independent runs.
>
> | Method            | ACC(%)                | F1(%)                   |
> |-------------------|-----------------------|-------------------------|
> | **BrainLM**       | 74.69 ± 3.39          | 73.89 ± 2.84            |
> | **STCAL**         | 77.78 ± 3.40          | 72.73 ± 1.97            |
> | **BrainMass**     | 80.86 ± 4.94          | 74.14 ± 2.99            |
> | **Brain-JEPA**    | 82.08 ± 2.62          | 77.37 ± 2.06            |
> | **BrainSymphony** | 83.33 ± 1.41          | 78.05 ± 4.17            |
> | **Starformer**    | 83.82 ± 2.59          | 81.08 ± 3.09            |
> | **Spike-DB**      | **84.09 ± 2.64***       | **85.12 ± 2.89***        |

---

> ### Comment · Reviewer_UykD · 2025-11-27
>
> I have increased my score to 6.

---

> > ### Author Response · Authors · 2025-11-27
> >
> > Thank you so much for acknowledging our response and for your decision to raise the rating! We sincerely appreciate your constructive feedback during the review process, which played a vital role in improving the quality of our work. Thank you again for your involvement and support.

---

### Meta-Review · Area_Chair_zs1G · 2026-01-04

**Summary:**

This paper proposes Spike-DB, a spike-based “digital brain” framework for modeling fMRI time series. It converts region-level BOLD signals into a spike representation, trains a spike transformer with an anchor-to-target prediction objective, and then uses the learned representations for downstream tasks including disease classification, abnormal region detection, and effective connectivity analysis. The reviewers’ main discussion focused on whether the spike encoding is biologically well-motivated for low-temporal-resolution fMRI, whether the evaluation was broad enough to support the headline claims, and whether the experimental details and reproducibility were sufficient. In the rebuttal, the authors clarified that the “spikes” are a biologically inspired region-level representation rather than neuron-level spiking, added variance/significance reporting, expanded experiments to a larger AMRD cohort, and provided more details on the downstream setup and computational cost. While one reviewer remains unconvinced, the added experiments and clarifications address most of the concrete concerns raised in the initial reviews and make the empirical claims more credible and easier to interpret.

**Reviewer Concerns:**

The rebuttal and revisions addressed many of the concrete issues raised in review. In particular, the authors clarified the intended biological interpretation of the spike representation as a region-level, HRF-inspired encoding rather than neuron-level spiking, softened overly strong novelty claims, and added missing experimental details. They substantially strengthened the empirical section by reporting variance and significance, adding ablations on the anchor/target setting, and expanding evaluation to a much larger external dataset (AMRD), which directly mitigates concerns about overfitting and limited scope. The downstream classification setup and computational cost are now better specified. Some conceptual discomfort remains about the use of “spikes” for fMRI and the realism of the most extreme masking setting, and the plan to release code only after acceptance is not ideal, but overall the main technical and empirical concerns that drove the lower scores were largely addressed.

**Reviewer Scores:**

If the reviewers had been able to update their scores after reading the authors’ responses, I would expect a modest but clear upward shift. The two reviewers who were initially at 4 have already indicated that they will move to 6, since most of their concerns were about missing justification, limited evaluation, and lack of statistical detail, all of which the authors addressed directly in their responses. The reviewers who were already at 6 would likely remain there, but with higher confidence. The single reviewer at 2 would probably not change their score, as their objections seem more fundamental and less tied to missing experimental detail. Overall, the score distribution would become more positive, with a clear majority above the acceptance threshold.

---

### Decision · Program_Chairs · 2026-01-26

Accept (Poster)